# Unlocking the Theory Behind Scaling 1-Bit Neural Networks

Majid Daliri[1], Zhao Song[2], Chiwun Yang[3]

[1]New York University, [2]Simons Institute for the Theory of Computing. UC Berkeley, [3]Sun Yat-sen University

daliri.majid@nyu.edu, magic.linuxkde@gmail.com, christiannyang37@gmail.com

Recently, 1-bit Large Language Models (LLMs) have emerged, showcasing an impressive combination of efficiency and performance that rivals traditional LLMs. Research by Wang et al. [1], Ma et al. [2] indicates that the performance of these 1-bit LLMs progressively improves as the number of parameters increases, hinting at the potential existence of a *Scaling Law in 1-bit Neural Networks*. This paper presents the **first theoretical result** that rigorously establishes this scaling law for 1-bit models. Our analysis starts with initializing a 1-bit two-layer linear network. We prove that, despite the constraint of weights restricted to $\{-1, +1\}$, its training dynamics inevitably align with kernel behavior as the network width grows. This theoretical breakthrough guarantees convergence of the 1-bit model to an arbitrarily small loss as width increases. Furthermore, we introduce the concept of the generalization difference, defined as the gap between the outputs of 1-bit networks and their full-precision counterparts, and demonstrate that this difference maintains a negligible level under the over-parameterization setting. Building on the work of Kaplan et al. [3], we examine how the training loss scales as a power-law function of the model size, dataset size, and computational resources utilized for training. Our findings underscore the promising potential of scaling 1-bit neural networks, suggesting that int1 could become the standard in future neural network precision.

## 1 Introduction

Large-scale neural networks, particularly Large Language Models (LLMs) [4, 5] and Large Multi-model Models (LMMs) [6, 7], are becoming increasingly relevant to our day-to-day lives, finding a huge variety of applications in both the workplace and at home [8, 9]. However, it is expensive to deploy and run these models due to their substantial computational requirements, large memory footprints, and energy consumption [10–12]. This is especially true for resource-constrained environments, such as mobile devices, edge computing, or companies with limited infrastructure [13–15]. To make these models more efficient and accessible, quantization techniques are used, which reduce the precision of the model's parameters (such as weights and activations) from floating-point numbers to lower-bit representations (e.g., 8-bit or even lower) [16–20]. Quantization reduces the memory and computational costs of inference, enabling faster processing with less energy, while maintaining a comparable level of performance. This optimization allows language models to be more practical, scalable, and sustainable for widespread use across various platforms [21–23].

In particular, quantization techniques could be primarily divided into two methods: Post-Training Quantization (PTQ) [24–26] and Quantization-Aware Training (QAT) [1, 2, 27]. PTQ methods, including uniform and non-uniform quantization, conveniently convert pre-trained model weights and activations to lower-bit representations post-training. However, this leads to accuracy loss, especially in lower precision, as the model is not optimized for these quantized representations and significant shifts in weight distribution occur [28]. The alternative, Quantization-Aware Training (QAT), incorporates quantization during training, allowing the model to fine-tune and adapt its parameters to the quantized representation, compensating for quantization errors. Therefore, compared to PTQ, QAT maintains higher accuracy and robustness even in lower precision.

Second Conference on Parsimony and Learning (CPAL 2025).

Recent studies [1, 2, 29, 30] have shown that 1-bit LLMs, most of which have matrix weights in the range of $\{-1, +1\}$, can be trained from scratch to deliver performance that rivals that of standard LLMs. These models exhibit remarkable efficiency, particularly in terms of scaling laws. Experimental results indicate that the performance of the 1-bit model improves as the number of parameters increases, a principle that mirrors the training approach utilized in standard LLMs [3]. Despite the demonstrated efficiency of quantization methods, our understanding of the training mechanism for quantization remains limited. Specifically, it remains unclear how and why the 1-bit QAT enhances learning capability as the number of neurons in the model is scaled up. In addition, we are also concerned about whether the quantization method damages the generalization ability compared to full precision networks.

In this study, we initially apply the Neural Tangent Kernel (NTK) framework to delve into the optimization and generalization issues associated with a two-layer linear network operating in 1-bit (int1) precision, as detailed in Section 4. We introduce a 1-bit quantization method to the hidden-layer weights $W \in \mathbb{R}^{d \times m}$ of the conventional NTK linear network, where $d$ represents the input dimension and $m$ indicates the model's width. Our analysis reveals that the training dynamics of the 1-bit model approximate kernel behavior as the model width $m$ expands. This key finding paves the way for an established relationship between the theoretically guaranteed loss and the model width, endowing the model with robust learning capabilities akin to kernel regression. Ultimately, the model achieves an insignificantly small training loss, contingent on setting a sufficiently large model width, selecting an appropriate learning rate, and allowing an adequate training duration.

Moreover, Section 5 provides a theoretical confirmation that, within the scaling trend, the disparities in predictions of the 1-bit model from those of the original linear network on identical inputs maintain a negligible value. We assess the error between our 1-bit linear and standard linear networks on both the training and test datasets. Our theorem demonstrates that for any input from these datasets, the absolute error between the two network predictions can be denoted as $\epsilon_{\text{quant}} \leq O(\kappa d \log(md/\delta))$ for scale coefficient $\kappa \leq 1$, model width $m$, dimension $d$ and failure probability $\delta \in (0, 0.1)$. This indicates that the output behavior of the 1-bit linear model increasingly aligns with that of the standard linear model. The observed similarity on the test dataset validates the generalization similarity, suggesting the feasibility of approximating training neural networks with int1 precision equivalent to full precision.

Finally, in Section 6, we verify our theoretical results by implementing training models to learn complicated functions to compare the difference between 1-bit networks and full precision networks. Firstly, we choose a combination of difficult functions across the exponential function, trigonometric function, logarithmic function, the Lambert W function, the Gamma function, and their combination. Therefore, we sample random data points and split train and test datasets. We next compare how the training loss decreases as the model width $m$ scales up. Besides, as shown in Section 6.3, in the trend of a growing number of parameters, the error of predictions both on training and test input likewise converge as the power-law in 1-bit networks optimization. In particular, we visualize some 1-dimension function to see how the differences of outputs are. We demonstrate the results complying with our theoretical guarantee with a negligible error.

## 2   Related Work

**Efficient Training Methods for Quantized Networks** Training large-scale neural networks with quantization introduces significant computational and memory savings, but it also presents challenges in optimization, particularly when dealing with extremely low precision formats like 1-bit or 8-bit. To address these challenges, several efficient training methods have been developed that aim to maintain accuracy while leveraging the benefits of quantization. One key method is Gradient Quantization, where the gradients during backpropagation are quantized to lower precision to reduce memory overhead and bandwidth during distributed training. Techniques like stochastic rounding are used to mitigate the impact of quantization noise, ensuring the training process remains stable and converges effectively.

Another important approach is Low-Rank Factorization [31, 32], which decomposes the large weight matrices in neural networks into smaller matrices, reducing the number of parameters that need to be updated during training. When combined with quantization, this method significantly reduces both the memory footprint and computational complexity, allowing for faster training on hardware with limited resources.

**Quantization Techniques for Accelerating Language Models** Beyond traditional weight and activation quantization, several advanced methods utilize quantization to enhance the efficiency of large language models (LLMs). One key approach is KV cache quantization [33–36], which reduces the memory footprint of transformer models during inference by quantizing the stored attention keys and values. This method is particularly beneficial for tasks involving long sequences, significantly speeding up inference and lowering memory consumption without a substantial loss in accuracy.

Another effective technique is mixed-precision quantization [37, 38], where different parts of the model are quantized at varying precision levels based on their sensitivity. For example, attention layers might use higher precision (e.g., 16-bit), while feedforward layers are quantized to 8-bit or lower. This balances computational efficiency and model performance. These strategies, combined with methods like activation pruning, showcase how targeted quantization can drastically accelerate LLMs while maintaining their effectiveness in real-world applications.

**Neural Tangent Kernel.** The study of Neural Tangent Kernel (NTK) [39] focuses on the gradient flow of neural networks during the training process, revealing that neural networks are equivalent to Gaussian processes at initialization in the infinite-width limit. This equivalence has been explored in numerous studies [40–54] that account for the robust performance and learning capabilities of over-parameterized neural networks. The kernel-based analysis framework provided by NTK is gaining popularity for its utility in elucidating the emerging abilities of large-scale neural networks. In a remarkable stride, Arora et al. [55] introduced the first exact algorithm for computing the Convolutional NTK (CNTK). This was followed by Alemohammad et al. [56] who proposed the Recurrent NTK, and Hron et al. [57] who presented the concept of infinite attention via NNGP and NTK for attention networks. These innovative works have showcased the enhanced performance achievable with the application of NTK to various neural network architectures. In a specific study, Malladi et al. [58] examined the training dynamics of fine-tuning Large Language Models (LLMs) using NTK, affirming the efficiency of such approaches.

## 3 Preliminary

In this section, we give the basic setups of this paper, which includes the introduction of the quantization method in this paper (Section 3.1), our NTK-style problem setup that we aim to solve in this paper (Section 3.2) and recalling the classical NTK setup for a two-layer linear network with ReLU activation function (Section 3.3).

### 3.1 Quantization

We first show how we reduce the computation of the inner product of two vectors from multiplication and addition operations to addition operations only, which is achieved by binarizing one of the vectors. This method could be extended to matrix multiplication easily since the basic matrix multiplication is to implement the inner product computation of two vectors in parallels. For a vector $w \in \mathbb{R}^d$, we define our quantization function as [1, 2]:

$$\mathrm{Quant}(w) := \mathsf{Sign}\Big(\mathsf{Ln}(w)\Big) \in \{-1, +1\}^d,$$

where $\mathsf{Ln}(w)$ is the normalization method that is given by: $\mathsf{Ln}(w) := \frac{w - E(w) \cdot \mathbf{1}_d}{\sqrt{V(w)}} \in \mathbb{R}^d$. Specially, we use $E(w) := \frac{1}{d} \sum_{k=1}^{d} w_k \in \mathbb{R}$ to denote the computational expectation of vector $w$ and use $V(w) := \|w - E(w) \cdot \mathbf{1}_d\|_2^2 \in \mathbb{R}$ to denote the corresponding variance.

Besides, the $k^{\text{th}}$ entry of signal function $\text{Sign}(z) \in \mathbb{R}^d$ for $z \in \mathbb{R}^d$, $k \in [d]$ is define by: $\text{Sign}_k(z) := \begin{cases} +1, & z_k \geq 0 \\ -1, & z_k < 0 \end{cases}$. Hence, we have a binary vector $\text{Quant}(w)$ where each entry of it is limited in the range $\{-1, +1\}$, and we denote that $\widetilde{w} := \text{Quant}(w)$ to simplify the notation. For any other vector $x \in \mathbb{R}^d$, addition operation $\sum_{k=1}^{d} \pm x_k$ is sufficient to compute $\langle \widetilde{w}, x \rangle$. After that, we introduce the dequantization function to recover the original computation result by showing:

$$\text{Dequant}(\langle \widetilde{w}, x \rangle) := \sqrt{V(w)} \cdot \langle \widetilde{w}, x \rangle + E(w) \cdot \langle \mathbf{1}, x \rangle.$$

## 3.2 NTK Problem Setup

**Data Points.** We consider a supervised learning task with a training dataset $\mathcal{D} = \{(x_i, y_i)\}_{i=1}^{n} \subset \mathbb{R}^d \times \mathbb{R}$, where each data point is under a mild assumption that $\|x_i\|_2 = 1$ and $y_i \leq 1$, $\forall i \in [n]$ [41]. Moreover, we are also concerned about the problem of the generalization of 1-bit models, we define the test dataset to compare 1-bit networks with standard networks, that is $\mathcal{D}_{\text{test}} := \{(x_{\text{test},i}, y_{\text{test},i})\}_{i=1}^{n} \subset \mathbb{R}^d \times \mathbb{R}$, where $\|x_{\text{test},i}\|_2 = 1$ and $y_{\text{test},i} \leq 1$, $\forall i \in [n]$.

**Model.** Here, we use hidden-layer weights $W = [w_1, w_2, \ldots, w_m] \in \mathbb{R}^{d \times m}$ and output-layer weights $a = [a_1, a_2, \ldots, a_m]^\top \in \mathbb{R}^m$. We consider a two-layer linear model $f$, which is defined as follows:

$$f(x, W, a) := \kappa \frac{1}{\sqrt{m}} \sum_{r=1}^{m} a_r \cdot \text{ReLU}\Big(\text{dq}(\langle \widetilde{w}_r, x \rangle)\Big),$$

where $\text{ReLU}(z) := \begin{cases} z, & z \geq 0 \\ 0, & z < 0 \end{cases}$, for all $z \in \mathbb{R}$, $\text{dq} : \mathbb{R} \to \mathbb{R}$ is a omitted version of dequantization function $\text{Dequant} : \mathbb{R} \to \mathbb{R}$, and $\widetilde{w}_r := \text{Quant}(w_r)$ as we denoted in previous section, $\kappa \in (0, 1]$ is a scale coefficient. Especially, we initialize each weight vector $w_r$, $\forall r \in [m]$ by sampling $w_r(0) \sim \mathcal{N}(0, \sigma \cdot I_d)$ with $\sigma = 1$. For output-layer $a$, we randomly sample $a_r \sim \text{Uniform}\{-1, +1\}$ independently for $r \in [m]$. Additionally, output-layer weight $a$ is fixed during the training.

**Training and Straight-Through Estimator (STE).** The training loss is measured by quadratic $\ell_2$ norm of the difference between model prediction $f(x_i, W, a)$ and ideal output vector $y_i$. Formally, we consider to train $W(t) = [w_1(t), w_2(t), \ldots, w_m(t)] \in \mathbb{R}^{d \times m}$ for $t \geq 0$ utilizing the following loss:

$$\mathsf{L}(t) := \frac{1}{2} \cdot \sum_{i=1}^{n} \|f(x_i, W(t), a) - y_i\|_2^2. \tag{1}$$

Moreover, since the signal function $\text{Sign}$ is not differentiable, we use Straight-Through Estimator (STE) to skip the signal function in back-propagation [1, 2, 59, 60], thus updating the trainable weights $W(t)$. For $t \geq 0$ and denote $\eta$ as the learning rate, we omit $f_i(t) := f(x_i, W(t), a) \in \mathbb{R}, \forall i \in [n]$, the formulation to update $r^{\text{th}}$ column of $W(t)$ for all $r \in [m]$ is given by:

$$w_r(t+1) := w_r(t) - \eta \sum_{i=1}^{n} (f_i(t) - y_i) \cdot \kappa a_r \mathbf{1}_{\text{dq}(\langle \widetilde{w}_r, x_i \rangle) \geq 0} x_i.$$

## 3.3 Recalling Classic NTK Setup

We now recall the classic NTK setup for the two-layer ReLU linear regression [61–64]. The function is given by: $f'(x, W, a) := \kappa \frac{1}{\sqrt{m}} \sum_{r=1}^{m} a_r \cdot \text{ReLU}\Big(\langle w_r, x \rangle\Big)$.

We define that $W'(0) := W(0) \in \mathbb{R}^{d \times m}$ to denote the trainable parameter for classic NTK setup, these two matrices are equal at initialization. For $t \geq 0$, we define the loss of training $f'$ as follows: $\mathsf{L}'(t) := \frac{1}{2} \cdot \sum_{i=1}^{n} \|f'(x_i, W'(t), a) - y_i\|_2^2$. Then the update of $W'(t)$ is: $W'(t+1) := W'(t) - \eta \cdot \nabla_{W'(t)} \mathsf{L}'(t)$.

# 4 Kernel Behavior and Training Convergence

We give our convergence analysis for training 1-bit model within the framework of Neural Tangent Kernel (NTK) in this section. First, we state our theoretical results that define the kernel function

in training and show how it converges to NTK and maintains the PD (Positive Definite) property in Section 4.1. Then we demonstrate the arbitrary small loss convergence guarantee of training 1-bit model (Eq. (1)) in Section 4.2. Finally, we give a general version of our theoretical scaling law analysis in Section 4.3.

## 4.1 Neural Tangent Kernel

Here, we utilize the NTK to describe the training dynamic of the 1-bit model. Following pre-conditions in the previous section, we define a kernel function, that denotes $H(t) \in \mathbb{R}^{n \times n}$ (Gram matrix). Especially, the $(i, j)$-th entry of $H(t)$ is given by:

$$H_{i,j}(t) := \kappa^2 \frac{1}{m} x_i^\top x_j \sum_{r=1}^{m} \mathbf{1}_{\mathsf{dq}(\langle \widetilde{w}_r(t), x_i \rangle) \geq 0} \mathbf{1}_{\mathsf{dq}(\langle \widetilde{w}_r(t), x_j \rangle) \geq 0}. \tag{2}$$

We define the formal NTK as $H^* := H(0) \in \mathbb{R}^{n \times n}$. Additionally, there's a commonly introduced assumption in NTK analysis: we denote the minimum value of eigenvalues of $A$ with $\lambda_{\min}(A)$ for any $A \in \mathbb{R}^{n \times n}$. In our work's context, we presuppose that $H$ is a Positive-definite (PD) matrix, meaning that $\lambda_{\min}(H^*) > 0$ [41].

**1-Bit ReLU Pattern.** The pattern of the Rectified Linear Unit (ReLU) function is determined by the indicator of function activation. As illustrated by Du et al. [41], in the settings of Section 3.3, the event $\mathbf{1}_{\langle w_r(0), x \rangle \geq 0} \neq \mathbf{1}_{\langle w, x \rangle \geq 0}$ happens infrequently for any $w, x \in \mathbb{R}^d$ that satisfies $\|w - w_r(0)\|_2 \leq R$. Notably, $R := \max_{r \in [m]} \|w_r(t) - w_r(0)\|_2 = \eta \| \sum_{\tau=1}^{t} \Delta w_r(\tau) \|_2$. In our analysis, for Eq. (2), the event $\mathbf{1}_{\mathsf{dq}(\langle \widetilde{w}_r(0), x \rangle) \geq 0} \neq \mathbf{1}_{\mathsf{dq}(\langle \widetilde{w}_r(t), x \rangle) \geq 0}$ is also unlikely to occur during training.

The convergence of $H(t)$ towards $H^*$, as well as the property of $H(t)$ being a PD matrix for any $t \geq 0$, can be validated by the following lemma:

**Lemma 4.1** (NTK convergence and PD property during the training, informal version of Lemma G.5). *Assume $\lambda_{\min}(H^*) > 0$. $\delta \in (0, 1)$, define $D := \max\{\sqrt{\log(md/\delta)}, 1\}$. Let $R \leq O(\lambda\delta/(\kappa^2 n^2 dD))$, then for any $t \geq 0$, with probability at least $1 - \delta$, we have: Part 1. $\|H(t) - H^*\|_F \leq O(\kappa^2 n^2 dRD/\delta)$. Part 2. $\lambda_{\min}(H(t)) \geq \lambda/2$.*

*Proof of Lemma 4.1.* The proof of Part 1 of this Lemma follows from the pattern $\mathbf{1}_{\mathsf{dq}(\langle \widetilde{w}_r(t), x_i \rangle) \geq 0}$ for $i \in [n]$ and $r \in [m]$ is rarely changed during the training, this habit is similar to the regular ReLU pattern $\mathbf{1}_{\langle w_r(t), x_i \rangle \geq 0}$ [41]. The proof of Part 2 of this Lemma can be obtained by plugging $R \leq O(\lambda\delta/(\kappa^2 n^2 dD))$. Please refer to Lemma G.5 for the detailed proof. □

## 4.2 Training Convergence

Having confirmed the convergence of the kernel function of the 1-bit linear network during training in Lemma 4.1, we can transform the dynamics of the loss function $\mathsf{L}(t)$ into the following **kernel behavior**:

$$\mathsf{L}(t+1) - \mathsf{L}(t) = -(\mathsf{F}(t) - y)^\top H(t)(\mathsf{F}(t) - y) + C_2 + C_3 + C_4$$
$$\approx -(\mathsf{F}(t) - y)^\top H(t)(\mathsf{F}(t) - y),$$

In this equation, $\mathsf{F}(t) = [f(x_1, W(t), a), \cdots, f(x_n, W(t), a)]^\top \in \mathbb{R}^n$ and $y = [y_1, \cdots, y_n]^\top \in \mathbb{R}^n$, while $C_2, C_3, C_4$ are negligible terms (please refer to Appendix I for a rigorous proof).

Further, by $\lambda_{\min}(H(t)) > 0$ (as per Part 2 of Lemma 4.1), for each optimization step $t \geq 0$, we find that $\mathsf{L}(t+1) \leq (1 - \eta\lambda/2)\mathsf{L}(t)$, thus ensuring a non-increase in loss. Given sufficient training iterations and an appropriately chosen learning rate, we can achieve training convergence, the confirmation of which is provided in the following section.

**Theorem 4.2** (Training convergence guarantee, informal version of Theorem I.1). *Given an expected error $\epsilon > 0$. Assume $\lambda_{\min}(H^*) > 0$. $\delta \in (0, 0.1)$, define $D := \sqrt{\log(md/\delta)}$. Choose $m \geq \Omega(\lambda^{-8} n^{12} d^8/(\delta\epsilon)^4)$, $\eta \leq O(\lambda\delta/(\kappa^2 n^2 dD))$. Then let $T \geq \Omega((\eta\lambda)^{-1} \log(ndD^2/\epsilon))$, with probability at least $1 - \delta$, we have: $\mathsf{L}(T) \leq \epsilon$.*

*Proof sketch of Theorem 4.2.* We first combine $\mathsf{L}(0) = O(\sqrt{n}dD^2)$ (Lemma I.3) and $\mathsf{L}(t + 1) \leq (1 - \eta\lambda/2)\mathsf{L}(t)$ (Lemma I.2), then we choose a sufficient large $T \geq \Omega((\eta\lambda)^{-1} \log(ndD^2/\epsilon))$ to achieve $\mathsf{L}(T) \leq \epsilon$. For the complete proof, please see Theorem I.1. $\qquad\square$

**Scaling Law for $1$-Bit Neural Networks.** Theorem 4.2 primarily illustrates a fact for any dataset with $n$ data points. After initializing the hidden-layer weights $W \in \mathbb{R}^{d\times m}$ from a normal distribution, and assuming the minimum eigenvalue of NTK $\lambda > 0$, we set $m$ to be a large enough value to ensure the network is sufficiently over-parameterized. With an appropriate learning rate, the loss can be minimized in finite training time to an arbitrarily small error $\epsilon$. This offers a crucial insight that confirms the existence of a *scaling law for 1-bit neural networks*, which is strictly bounded by the model width $m$ and training steps $T$. Consequently, we present the following Proposition that elucidates the principle of training 1-bit linear networks from scratch. This proposition is built upon Theorem 4.2 and the principle of training loss that scales as a power-law with model size, dataset size, and the amount of compute used for training [3, 65].

**Proposition 4.3** (Scaling Law for 1-Bit Neural Networks)**.** $\delta \in (0, 0.1)$. *Define* $\mathsf{N} := O(md)$ *as the number of parameters,* $\mathsf{D} := O(n)$ *as the size of training dataset,* $\mathsf{C} := O(\mathsf{N}\mathsf{D}T)$ *as the total compute cost. Especially, we denote the scale coefficients as* $\alpha := \mathsf{D}d\log(md/\delta)$, *and we then choose* $\eta \leq O(\lambda\delta/(m\kappa^2 n^2 dD))$ *and* $T \geq \Omega((\eta\lambda m)^{-1}\log(nd\log(md/\delta)/\epsilon))$. *Thus, the training loss, denoted as* $\mathsf{L}_{\mathrm{scale}}$, *satisfies:*

$$\mathsf{L}_{\mathrm{scale}} \approx \max\{\frac{\mathsf{D}^3 \cdot d^{2.25}}{\lambda^2 \mathsf{N}^{0.25}}, \frac{\alpha}{\exp(\eta\lambda\mathsf{C})}\}.$$

*Proof of Proposition 4.3.* This proof follows from the definitions of $\mathsf{N}$, $\mathsf{D}$, $\mathsf{C}$ and $\alpha$. Then, by choosing $\eta \leq O(\lambda\delta/(mn^2 dD))$ and $T \geq \Omega((\eta\lambda m)^{-1}\log(nd\log(md/\delta)/\epsilon))$, we utilize Theorem 4.2 to obtain our proposition. $\qquad\square$

Proposition 4.3 demonstrates that the training loss of the prefix learning converges exponentially as we increase the computational cost $\mathsf{C}$, which primarily depends on the number of parameters and the training time in prefix learning. This further suggests a potential relationship for formulating a scaling law for 1-bit neural networks.

## 4.3 Extensibility

We now bridge our theoretical framework to a real-world application involving a multi-layer 1-bit transformer trained on large-scale datasets. Let the full dataset be denoted as $\mathcal{D}_{\mathrm{mat}} = \{(X_i, Y_i)\}_{i=1}^n \subset \mathbb{R}^{K\times d}$, where $X_i \in \mathbb{R}^{K\times d}$ represents a sequence of $K$ tokens with $d$-dimensional embeddings, and $Y_i$ denotes the corresponding target sequence. Here, $K$ is the input context length.

The standard transformer architecture [10] interleaves multi-head self-attention and position-wise feed-forward layers. For an input sequence $X \in \mathbb{R}^{K\times d}$ (compactly representing $K$ token embeddings), an $N$-layer transformer is defined recursively as:

$$\mathcal{F}(X) := \mathsf{TF}_{(N)}\left(\mathsf{TF}_{(N-1)}\left(\cdots\mathsf{TF}_{(1)}(X + E)\cdots\right)\right),$$

where $E \in \mathbb{R}^{K\times d}$ is the positional embedding matrix, and $\mathsf{TF}_{(\nu)} : \mathbb{R}^{K\times d} \to \mathbb{R}^{K\times d}$ for $\nu \in [N]$ denotes the $\nu$-th transformer block. For brevity, we omit layer indices when describing a single transformer block $\mathsf{TF}$, which consists of:

$$\mathsf{Attn}(X) := X + \sum_{\xi=1}^h \mathsf{dq}\left(\mathsf{softmax}\left(\frac{\mathsf{dq}(X\widetilde{W}_{\xi,Q}\widetilde{W}_{\xi,K}^\top X^\top)}{\sqrt{d}}\right)X\widetilde{W}_{\xi,V}\widetilde{W}_{\xi,O}^\top\right),$$

$$\mathsf{FF}(X) := X + \mathsf{dq}\left(\mathsf{ReLU}\left(\mathsf{dq}(X\widetilde{W}_1) + \mathbf{1}_K b_1^\top\right)\widetilde{W}_2^\top\right) + \mathbf{1}_K b_2^\top,$$

$$\mathsf{TF}(X) := \mathsf{FF}(\mathsf{Attn}(X)),$$

where: - $h$ is the number of attention heads. - $\widetilde{W}$ denotes 1-bit quantized weights, with $\mathsf{dq}(\cdot)$ as the dequantization operator. - For each head $\xi \in [h]$, $\widetilde{W}_{\xi,Q}, \widetilde{W}_{\xi,K}, \widetilde{W}_{\xi,V} \in \mathbb{R}^{d\times d'}$ and $\widetilde{W}_{\xi,O} \in \mathbb{R}^{d'\times d}$

are query, key, value, and output projection matrices, respectively. - In the feed-forward network, $\widetilde{W}_1 \in \mathbb{R}^{d \times m}$ and $\widetilde{W}_2 \in \mathbb{R}^{m \times d}$ are projection matrices, with $m$ as the hidden dimension, while $b_1 \in \mathbb{R}^m$ and $b_2 \in \mathbb{R}^d$ are bias terms.

The full parameters of the model is denoted as $\theta_{(d',h,m)} := \{W_{\nu,\xi,Q}, W_{\nu,\xi,K}, W_{\nu,\xi,V}, W_{\nu,\xi,O},$ $W_{\nu,2}, W_{\nu,2}, b_{\nu,1}, b_{\nu,2}\}_{(\nu,\xi) \in [L] \times [h]} + \{E\}$. Given a loss metric $\ell(\widehat{Y}, Y) := \frac{1}{2}\|\widehat{Y} - Y\|_F^2$, we define the training objective as follows:

$$\mathcal{L}(\theta_{(d',h,m)}) := \sum_{i=1}^{n} \ell(\mathcal{F}(X_i), Y_i). \tag{3}$$

Thus, we establish a general version of our theory:

**Proposition 4.4.** *Given an expected error $\epsilon > 0$ and denote the failure probability $\delta \in (0, 0.1)$. Given a dataset $\mathcal{D}_{\mathrm{mat}} = \{(X_i, Y_i)\}_{i=1}^{n} \subset \mathbb{R}^{K \times d}$ and a model function $\mathcal{F} : \mathbb{R}^{K \times d} \to \mathbb{R}^{K \times d}$ with parameters set $\theta_{(d',h,m)}$. Assuming each NTK of $\mathcal{F}$ is PD, denoted $H_{k,j}^*$ for $(k,j) \in [K] \times [d]$, $\lambda_{\min}(H_{k,j}^*) > 0$. Define $\lambda := \min_{(k,j) \in [K] \times [d]}\{\lambda_{\min}(H_{k,j}^*)\}$, we choose $m \geq \Omega(\lambda^{-8} n^{12} K^{12} d^{20}/(\delta\epsilon)^4)$. Then with a probability at least $1 - \delta$, there exists at least one first-order algorithm that minimizes Eq. (3) to $\epsilon$.*

*Proof.* We consider a special case that only optimizes one feed-forward layer of the model, then solving $\mathcal{L}(\theta_{(d',h,m)})$ is just letting $n = Kdn'$ where $n'$ represents the data size in Theorem 4.2. $\square$

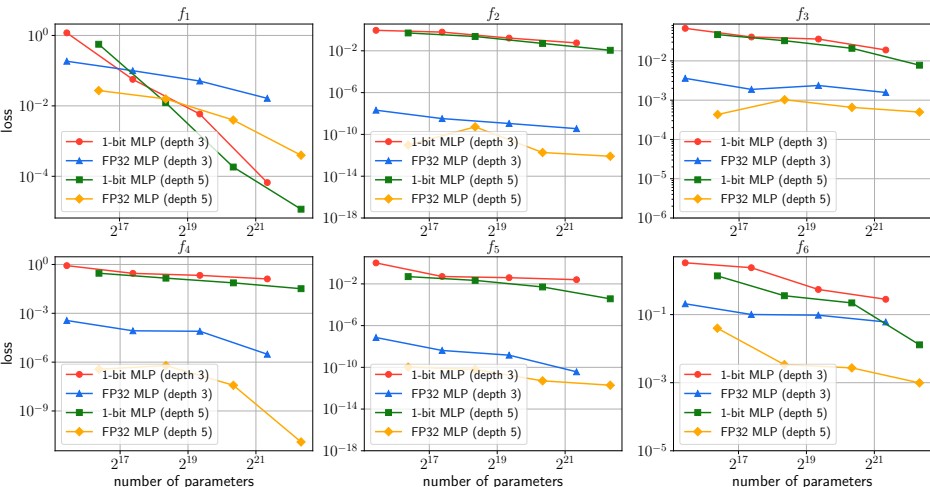

Figure 1: Verification experiment for *scaling law for* 1-*bit neural networks*. Minimum training loss of scaling number of parameters for MLP model to learn complicated functions $f_1, f_2, f_3, f_4, f_5$ and $f_6$, and these function is defined in Section 6.1.

# 5 Generalization Similarity

In this section, we present our theoretical analysis that proves that training large-scale 1-bit neural networks is equivalent to training standard large-scale neural networks. In Section 5.1, we explain how the difference between the outputs of our 1-bit model and outputs of the standard NTK-style linear network for the same input at initialization, which is defined as function difference at initialization, will be kept in a small error while the model width (denoted as $m$) increase. Next, in Section 5.2, we confirm that in the trend of scaling up the model width, during the training, the predictions of 1-bit model and full precision model are also similar to a very slight error on both the training dataset and the test dataset.

## 5.1 Function Difference at Initialization

To begin with, at initialization, the boundary on $|f(x, W(0), a) - f'(x, W'(0), a)|$ is stated as follows:

**Lemma 5.1** (Function difference at initialization, informal version of Lemma K.4). $\delta \in (0, 0.1)$. *Denote* $D := \sqrt{\log(md/\delta)}$. $\forall x \in \mathbb{R}^d$ *that satisfies* $\|x\|_2 = 1$, *for any initial quantization error* $\epsilon_{\text{init}} > 0$, *we choose* $\kappa \leq O(\epsilon_{\text{init}}/(\sqrt{d}D^2))$. *Then with a probability* $1 - \delta$, *we have:* $|f(x, W(0), a) - f'(x, W'(0), a)| \leq \epsilon_{\text{init}}$.

*Proof sketch of Lemma 5.1.* Due to the initialization of $W(0)$ and $W'(0)$, we then have the tail bound of the Gaussian distribution. Hence, the difference could be bounded by Hoeffding bound, we then get the result. Please refer to Lemma K.4 for the formal proof of this Lemma. $\qquad\square$

## 5.2 Generalization Similarity

We now address whether using 1-bit precision compromises the generalization ability of standard neural networks. Specifically, we use the test dataset to evaluate the **generalization similarity** - a measure of the similarity between two functions on out-of-distribution (OOD) data. This measure is designed to assess the equivalence between two functions. If, during each step of training two networks, these two training processes are deemed equivalent, then we assert that the generalization similarity is valid.

Addressing the above concern, we demonstrate that the predictions of two functions on both training and test datasets can be bounded to an arbitrarily small quantization error, provided that $m$ is sufficiently large. Theoretically, as $m$ scales towards infinity, the quantization error converges to 0. This finding confirms the validity of our generalization similarity measure and asserts that 1-bit precision does not compromise the generalization ability of standard neural networks.

**Theorem 5.2** (Training and generalization similarity, informal version of Theorem K.1). *Let all preconditions in Theorem 4.2 satisfy. For any quantization error* $\epsilon_{\text{quant}} > 0$, *we choose* $\kappa \leq O(\epsilon_{\text{quant}}/(dD^2))$. *Integer* $\forall t \geq 0$. *For any training input* $x_i \in \mathbb{R}^d$ *in* $\mathcal{D}$ *and any test input* $x_{\text{test},i} \in \mathbb{R}^d$ *in* $\mathcal{D}_{\text{test}}$, *with a probability at least* $1 - \delta$, *we have:*

- *Part 1.* $|f(x_i, W(t), a) - f(x_i, W(t), a)| \leq \epsilon_{\text{quant}}$.

- *Part 2.* $|f(x_{\text{test},i}, W(t), a) - f(x_{\text{test},i}, W(t), a)| \leq \epsilon_{\text{quant}}$.

*Proof.* Proof sketch of Theorem 5.2 Since we proved $|f(x, W(0), a) - f'(x, W'(0), a)| \leq \epsilon_{\text{init}}$ in Lemma 5.1, then as we choose appropriate $R$ and learning rate $\eta$, the equations in Part 1 and Part 2 of this Theorem would be bounded by scaling $m$ to be sufficiently large. We state the complete proof in Theorem K.1. $\qquad\square$

**Training Equivalence.** Here, we say training $f$ and $f'$ are equivalent since we achieve the predictions that these two functions are extremely similar by plugging an appropriate value of $\kappa$. Besides, as we proved in Theorem 4.2, this implementation would not harm the optimization of 1-bit networks. This further explains why 1-bit precision even processes better when the scales of networks are increasing, instead of turning to a training collapse. Therefore, we believe it is the theory unlocking the potential of 1-bit neural networks from the perspective of kernel-based analysis.

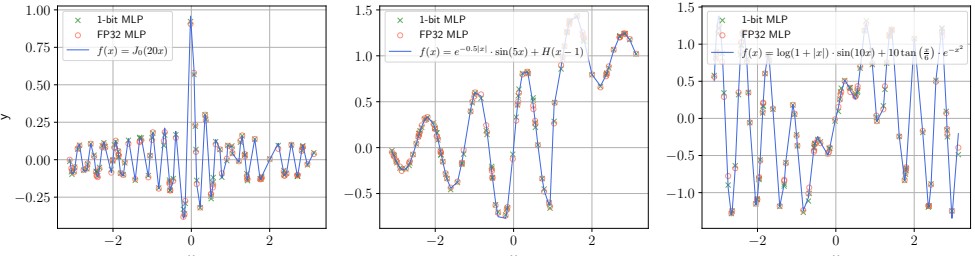

Figure 2: This plot shows the difference between the predicted and actual values of the functions on the test dataset. We tested three complex functions, as seen in the images, and the performance of the 1-bit model is nearly identical to that of the standard 32-bit floating-point model.

# 6 Experiments

In this section, we aim to verify our theory by evaluating how well our quantization works for learning rigorous functions and comparing it to the standard model. We designed our experiment to 1) validate the scaling law (Section 6.1), 2) visually demonstrate that the performance difference is minimal compared to the standard model, which uses full-bit precision, through visualizations of single-variable input functions (Section 6.2), and 3) show how the test and train losses decrease as the model's parameter size increases and as the epochs progress (Section 6.3).

## 6.1 Verification on Scaling Law

**Experiment Setup.** In this experiment, we aimed to learn rigorous functions using a Multi-Layer Perceptron (MLP) with varying depths of 3 and 5 layers. The MLP models had different sizes for the hidden layers, and we measured the minimum loss achieved throughout the training process. Each model was trained for 100,000 steps. We experimented with various parameter sizes and plotted the corresponding loss functions. Additionally, we compared our method with the standard training approach using 32-bit floating-point precision.

We experimented with a variety of target functions, and for each function, the inputs $x_i$ were randomly chosen within the range $[-1, 1]$. Specifically, each $x_i$ was sampled from a uniform distribution over this interval to ensure that the network could handle input values across the entire domain of interest. We sampled 100 data points and trained our model over this set.

The functions we aimed to learn during the experiment are listed below:

1. $f_1(x_1, x_2, x_3, x_4, x_5) = \exp\left(\frac{1}{5}\sum_{i=1}^{5}\sin^2\left(\frac{\pi x_i}{2}\right)\right)$, This function takes five inputs and applies a sinusoidal transformation followed by an exponential operation.

2. $f_2(x_1, x_2, x_3, x_4) = \ln(1+|x_1|) + \left(x_2^2 - x_2\right) + \sin(x_3) - e^{x_4}$, the function combines logarithmic, polynomial, trigonometric, and exponential components over four input variables.

3. $f_3(x_1, x_2, x_3) = x_1 \times x_2 - x_3$, This is a simple linear function over three inputs, involving multiplication and subtraction.

4. $f_4(x_1, x_2, x_3, x_4) = x_0 \cdot \sin(x_1) + \cos(x_2) - 0.5 \cdot x_3$, A four-input function mixing trigonometric and linear terms, with coefficients applied to the terms.

5. $f_5(x_1, x_2, x_3, x_4) = \frac{x_0^2}{1+|x_1|} - e^{x_2} + \tanh(x_3) + \sqrt{|x_0 \cdot x_2|}$, This function incorporates nonlinear operations like exponentials, hyperbolic tangents, and square roots.

6. $f_6(x_1, x_2, x_3, x_4) = \text{LambertW}(x_0 \cdot x_1) + \frac{x_2}{\log(1+e^{x_3})} - \frac{\Gamma(x_1)}{1+|x_0|}$, The most complex function we tested, which includes special functions like the Lambert W function and the Gamma function, alongside logarithmic and exponential components.

We compare our quantized model (using INT1, $32\times$ smaller) to a standard non-quantized model (using 32-bit precision). For all functions ($f_1$ to $f_6$), we observe (in ) that as the number of parameters increases, the loss decreases, supporting our theoretical claim that larger models lead to convergence.

Although the standard method generally performs better due to its 32-bit precision, the gap decreases as the number of parameters grows. This shows that while our method has a slightly higher loss, it remains competitive, offering significant memory and computational efficiency.

## 6.2 Comparison on 1-D Functions

In this experiment, we aimed to visually demonstrate the performance on highly complex functions with sharp spikes between $[-\pi, \pi]$. We sampled 100 uniformly spaced points and trained a 2-layer MLP with 20M parameters to learn the function. Additionally, we sampled 100 random points uniformly from this interval as the test dataset.

The first observation from the plot is that both the standard and 1-bit methods learn all the functions almost perfectly, with minimal difference between them. Secondly, both methods perform similarly

on these functions, which can be easily observed by comparing the scatter plots of the 1-bit and standard models. The 1-bit model requires $32\times$ less energy and computation.

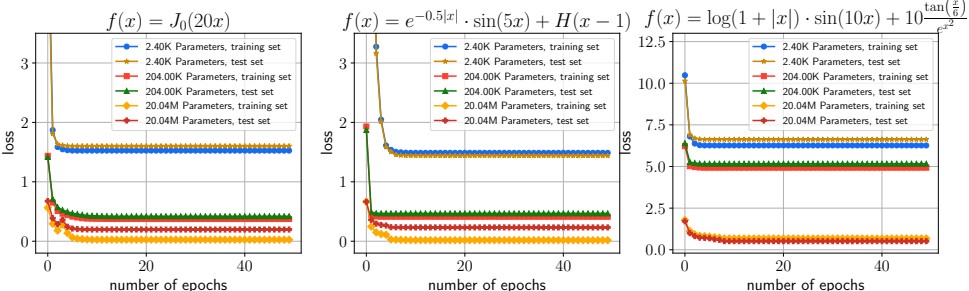

Figure 3: This plot shows the $\ell_2$ difference between both the training and test points and the predicted points throughout the training phase for different model sizes and parameter counts. Each plot demonstrates how the error decreases as training progresses, highlighting the impact of model size on both training and test performance.

## 6.3 Evaluation on Training and Generalization Similarity

For the same set of functions, we show how the loss functions for both the train and test datasets decrease as the number of epochs increases. As the training progresses, the loss converges towards zero for models with a higher number of parameters. We experimented with models containing 2.4k, 204k, and 20M parameters, each consisting of only 2 layers.

Across all three functions, the loss decreases rapidly in the early epochs and stabilizes for both the training and test sets. Larger models with 20M parameters consistently achieve lower final losses compared to smaller models with 2.4k and 204k parameters, demonstrating the benefit of increased model size. The gap between training and test losses remains minimal, indicating strong generalization across different parameter sizes. More importantly, the key observation is that the models predict similarly on both the training and test datasets, a behavior we refer to as *generalization similarity*. This means that the models, regardless of size, behave similarly across both datasets, supporting the scaling law that increasing model size leads to better convergence and generalization, but also highlighting the consistent similarity in performance between training and testing across different functions.

# 7 Conclusion

In conclusion, our theoretical results confirm the scaling law for 1-bit neural networks. We demonstrated that the model achieves a small loss as the number of parameters increases. Despite the constraint of binary weights, 1-bit models show similar behavior to full-precision models as their width grows. Our experiments support this theory, showing that 1-bit networks perform nearly as well as standard models on complex functions. As the number of parameters grows, the performance gap between 1-bit and full-precision models reduces. These findings highlight that 1-bit networks are both efficient and effective, providing a strong alternative to traditional models.

## Acknowledgement

The authors sincerely thank Bo Chen, Xiaoyu Li, Zhizhou Sha, Jing Xiong, Junwei Yu and Yufa Zhou for their helpful suggestions and discussion. The authors also would like to thank all anonymous reviewers for their constructive reviews that enhanced the contribution of this work.

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

# Appendix

## Contents

# Roadmap

We initially introduce the intention of each section in the appendix here. In Appendix A, we review more prior works that relate to our work. In Appendix B, we provide the preliminary for our theoretical analysis. In Appendix C, we give the formal definition of the NTK-style problem setup we aim to solve in this paper. In Appendix D, we strictly define the quantization method we utilize for our approach. We discuss the potential pattern changing of ReLU and signal function in Appendix E. For optimizing 1-bit neural network, we state the Straight-Through Estimator method (STE) definitions in Appendix F. In Appendix G, we define NTK for our optimization problem and discuss its properties. In Appendix I, we prove the convergence guarantee of training 1-bit neural networks. In Appendix J, we review the classical setup of solving the NTK-style linear regression. We confirm the generalization similarity in Appendix K.

# A  More Related Work

**Theoretical Approach for Understanding Modern Neural Networks.** The intricate architecture of transformer-based models, coupled with the stochastic nature of their optimization processes, presents a formidable challenge in comprehending the behaviors of large language models (LLMs). However, delving into these complexities through a theoretical lens can illuminate pathways for enhancing and innovating future AI systems. This exploration encompasses various facets, including the **optimization strategies for LLMs** [52, 66, 67], the intricacies of **white-box transformers** [68–71], and the analysis of **emergent capabilities** that arise within these models [4, 72–76]. Additionally, the **modern Hopfield model** [77–83] offers a rich terrain for investigation, revealing the nuanced dynamics that govern these advanced neural networks.

**Efficient Neural Networks.** As the principles of scaling laws come to the forefront, contemporary neural networks are increasingly trained on expansive datasets, necessitating substantial computational resources [11, 84–94]. This demand for efficiency has spurred research into algorithms that optimize **computational complexity**, minimize **memory usage**, and enhance **alignment with GPU architectures**. Such advancements are crucial in navigating the challenges posed by the ever-growing scale of data and the intricate demands of modern AI applications, ensuring that these powerful tools remain accessible and effective in their deployment.

# B  Preliminary

## B.1  Notations

In this paper, we use integer $m > 0$ to denote the width of neural networks, in particular, $m$ is sufficiently large. We use integer $d > 0$ to denote the dimension of neural networks. We use integer $n > 0$ to denote the size of the training dataset.

## B.2  Basic Facts

**Fact B.1.** *For a variable $x \sim \mathcal{N}(0, \sigma^2)$, then with probability at least $1 - \delta$, we have:*
$$|x| \le C\sigma\sqrt{\log(1/\delta)}$$

**Fact B.2.** *For an 1-Lipschitz function $f(\cdot)$, we have:*
$$|f(x) - f(y)| \le |x - y|, \forall x, y \in \mathbb{R}^d$$

**Fact B.3.** *For a Gaussian variable $x \sim \mathcal{N}(0, \sigma^2 \cdot I_d)$ where $\sigma \in \mathbb{R}$, then for any $t > 0$, we have:*
$$\Pr[x \le t] \le \frac{2t}{\sqrt{2\pi}\sigma}$$

**Fact B.4.** *For a Gaussian vector $w \sim \mathcal{N}(0, \sigma^2 \cdot I_d)$ where $\sigma \in \mathbb{R}$, and a fixed vector $x \in \mathbb{R}^d$, we have:*
$$w^\top x \sim \mathcal{N}(0, \sigma^2\|x\|_2^2 \cdot I_d)$$

**Fact B.5.** *For two matrices $H, \widetilde{H} \in \mathbb{R}^{n \times n}$, we have:*

$$\lambda_{\min}(\widetilde{H}) \geq \lambda_{\min}(H) - \|H - \widetilde{H}\|_F$$

**Fact B.6.** *For $x \in (0, 1)$, integer $t \geq 0$, we have:*

$$\sum_{\tau=1}^{t} (1-x)^{\tau} \leq -\frac{1}{\log(1-x)} \leq \frac{2}{x}$$

## B.3 Probability Tools

Here, we state a probability toolkit in the following, including several helpful lemmas we'd like to use. Firstly, we provide the lemma about Chernoff bound in [95] below.

**Lemma B.7** (Chernoff bound, [95]). *Let $X = \sum_{i=1}^{n} X_i$, where $X_i = 1$ with probability $p_i$ and $X_i = 0$ with probability $1 - p_i$, and all $X_i$ are independent. Let $\mu = \mathbb{E}[X] = \sum_{i=1}^{n} p_i$. Then*

- $\Pr[X \geq (1+\delta)\mu] \leq \exp(-\delta^2\mu/3), \forall \delta > 0$;

- $\Pr[X \leq (1-\delta)\mu] \leq \exp(-\delta^2\mu/1), \forall 0 < \delta < 1$.

Next, we offer the lemma about Hoeffding bound as in [96].

**Lemma B.8** (Hoeffding bound, [96]). *Let $X_1, \cdots, X_n$ denote $n$ independent bounded variables in $[a_i, b_i]$ for $a_i, b_i \in \mathbb{R}$. Let $X := \sum_{i=1}^{n} X_i$, then we have*

$$\Pr[|X - \mathbb{E}[X]| \geq t] \leq 2\exp(-\frac{2t^2}{\sum_{i=1}^{n}(b_i - a_i)^2})$$

We show the lemma of Bernstein inequality as [97].

**Lemma B.9** (Bernstein inequality, [97]). *Let $X_1, \cdots, X_n$ denote $n$ independent zero-mean random variables. Suppose $|X_i| \leq M$ almost surely for all $i$. Then, for all positive $t$,*

$$\Pr[\sum_{i=1}^{n} X_i \geq t] \leq \exp(-\frac{t^2/2}{\sum_{j=1}^{n} \mathbb{E}[X_j^2] + Mt/3})$$

Then, we give the Khintchine's inequality in [98, 99] as follows:

**Lemma B.10** (Khintchine's inequality, [98, 99]). *Let $\sigma_1, \cdots, \sigma_n$ be i.i.d sign random variables, and let $z_1 \cdots, z_n$ be real numbers. Then there are constants $C > 0$ so that for all $t > 0$*

$$\Pr[|\sum_{i=1}^{n} z_i \sigma_i| \geq t\|z\|_2] \leq \exp(-Ct^2)$$

We give Hason-wright inequality from [100, 101] below.

**Lemma B.11** (Hason-wright inequality, [100, 101]). *Let $x \in \mathbb{R}^n$ denote a random vector with independent entries $x_i$ with $\mathbb{E}[x_i] = 0$ and $|x_i| \leq K$ Let $A$ be an $n \times n$ matrix. Then, for every $t \geq 0$*

$$\Pr[|x^\top A x - \mathbb{E}[x^\top A x]| > t] \leq 2\exp(-c\min\{t^2/(K^4\|A\|_F^2), t/(K^2\|A\|)\})$$

We state Lemma 1 on page 1325 of Laurent and Massart [102].

**Lemma B.12** (Lemma 1 on page 1325 of Laurent and Massart, [102]). *Let $X \sim \mathcal{X}_k^2$ be a chi-squared distributed random variable with $k$ degrees of freedom. Each one has zero mean and $\sigma^2$ variance. Then*

$$\Pr[X - k\sigma^2 \geq (2\sqrt{kt} + 2t)\sigma^2] \leq \exp(-t)$$
$$\Pr[X - k\sigma^2 \geq 2\sqrt{kt}\sigma^2] \leq \exp(-t)$$

Here, we provide a tail bound for sub-exponential distribution [103].

**Lemma B.13** (Tail bound for sub-exponential distribution, [103]). *We say $X \in \mathrm{SE}(\sigma^2, \alpha)$ with parameters $\sigma > 0, \alpha > 0$, if*

$$\mathbb{E}[e^{\lambda X}] \leq \exp(\lambda^2 \sigma^2 / 2), \forall |\lambda| < 1/\alpha.$$

*Let $X \in \mathrm{SE}(\sigma^2, \alpha)$ and $\mathbb{E}[X] = \mu$, then:*

$$\Pr[|X - \mu| \geq t] \leq \exp(-0.5 \min\{t^2/\sigma^2, t/\alpha\})$$

In the following, we show the helpful lemma of matrix Chernoff bound as in [104, 105].

**Lemma B.14** (Matrix Chernoff bound, [104, 105]). *Let $\mathcal{X}$ be a finite set of positive-semidefinite matrices with dimension $d \times d$, and suppose that*

$$\max_{X \in \mathcal{X}} \lambda_{\max}(X) \leq B.$$

*Sample $\{X_1, \cdots, X_n\}$ uniformly at random from $\mathcal{X}$ without replacement. We define $\mu_{\min}$ and $\mu_{\max}$ as follows:*

$$\mu_{\min} := n \cdot \lambda_{\min}(\mathop{\mathbb{E}}_{X \in \mathcal{X}}(X))$$

$$\mu_{\max} := n \cdot \lambda_{\max}(\mathop{\mathbb{E}}_{X \in \mathcal{X}}(X)).$$

*Then*

$$\Pr[\lambda_{\min}(\sum_{i=1}^{n} X_i) \leq (1 - \delta)\mu_{\min}] \leq d \cdot \exp(-\delta^2 \mu_{\min}/B) \text{ for } \delta \in (0, 1],$$

$$\Pr[\lambda_{\max}(\sum_{i=1}^{n} X_i) \geq (1 + \delta)\mu_{\max}] \leq d \cdot \exp(-\delta^2 \mu_{\max}/(4B)) \text{ for } \delta \geq 0.$$

Finally, we state Markov's inequality as below.

**Lemma B.15** (Markov's inequality). *If $X$ is a non-negative random variable and $a > 0$, then the probability that $X$ is at least $a$ is at most the expectation of $X$ divided by $a$:*

$$\Pr[X \geq a] \leq \frac{\mathbb{E}[X]}{a}$$

## B.4 Basic Bound

**Definition B.16.** *For $\delta \in (0, 0.1)$ and a sufficiently large constant $C > 0$, we define:*

$$D := \max\{C\sqrt{\log(md/\delta)}, 1\}$$

# C NTK Problem Setup

## C.1 Dataset

We consider a dataset where each data point is a tuple that includes a vector input and a scalar output. In particular, we assume that $\ell_2$ norm of each input equals $1$ and the absolute value of each target is not bigger than $1$. We give the formal definition as follows:

**Definition C.1** (Data Points). *We define dataset $\mathcal{D} := \{(x_i, y_i)\}_{i=1}^{n} \subset \mathbb{R}^d \times \mathbb{R}$, where $\|x_i\|_2 = 1$ and $|y_i| \leq 1$ for any $i \in [n]$.*

## C.2 Model

**Weights and Initialization.**

**Definition C.2.** *We give the following definitions:*

- **Hidden-layer weights $W \in \mathbb{R}^{d \times m}$.** *We define the hidden-layer weights $W := [w_1, w_2, \cdots, w_m] \in \mathbb{R}^{d \times m}$ where $w_r \in \mathbb{R}^d, \forall r \in [m]$.*

- **Output-layer weights** $a \in \mathbb{R}^m$. We define the output-layer weights $a := [a_1, a_2, \cdots, a_m]^\top \in \mathbb{R}^m$, especially, vector $a$ is fixed during the training.

**Definition C.3.** *We give the following initializations:*

- **Initialization of hidden-layer weights** $W \in \mathbb{R}^{d \times m}$. *We randomly initialize* $W(0) := [w_1(0), w_2(0), \cdots, w_m(0)] \in \mathbb{R}^{d \times m}$, *where its $r$-th column for $r \in [m]$ is sampled by* $w_r(0) \sim \mathcal{N}(0, \sigma^2 \cdot I_d)$ *with* $\sigma^2 = 1$.

- **Initialization of output-layer weights** $a \in \mathbb{R}^m$. *We randomly initialize $a \in \mathbb{R}^m$ where its $r$-th entry for $r \in [m]$ is sampled by* $a_r \sim \mathsf{Uniorm}\{-1, +1\}$.

**Model.**

**Definition C.4.** *For a scalar $x \in \mathbb{R}$, we define:*

$$\mathsf{ReLU}(x) = \max\{0, x\} \in \mathbb{R}$$

**Definition C.5.** *If the following conditions hold:*

- *For a input vector $x \in \mathbb{R}^d$.*

- *For a hidden-layer weights $W \in \mathbb{R}^{d \times m}$ as Definition C.2.*

- *For a output-layer weights $a \in \mathbb{R}^m$ as Definition C.2.*

- *Let $\mathsf{q} : \mathbb{R}^d \to \{-1, +1\}^d$ be defined as Definition D.4.*

- *Let $\mathsf{dq} : \mathbb{R} \to \mathbb{R}$ be defined as Definition D.5.*

- *Denote $\widetilde{w}_r = \mathsf{q}(w_r) \in \{-1, +1\}^d$.*

- *Let $\mathsf{ReLU} : \mathbb{R} \to \mathbb{R}$ be defined as Definition C.4.*

- *For $\kappa \in (0, 1]$.*

*We define:*

$$f(x, W, a) := \kappa \frac{1}{\sqrt{m}} \sum_{r=1}^m a_r \cdot \mathsf{ReLU}\Big(\mathsf{dq}(\langle \widetilde{w}_r, x \rangle)\Big) \in \mathbb{R}$$

**Lemma C.6.** *If the following conditions hold:*

- *For a input vector $x \in \mathbb{R}^d$.*

- *For a hidden-layer weights $W \in \mathbb{R}^{d \times m}$ as Definition C.2.*

- *For a output-layer weights $a \in \mathbb{R}^m$ as Definition C.2.*

- *Let $\mathsf{q} : \mathbb{R}^d \to \{-1, +1\}^d$ be defined as Definition D.4.*

- *Let $\mathsf{dq} : \mathbb{R} \to \mathbb{R}$ be defined as Definition D.5.*

- *Denote $\widetilde{w}_r = \mathsf{q}(w_r) \in \{-1, +1\}^d$.*

- *Let $\mathsf{ReLU} : \mathbb{R} \to \mathbb{R}$ be defined as Definition C.4.*

- *Let $u : \mathbb{R}^d \to \mathbb{R}^d$ be defined as Definition D.6.*

- *For $\kappa \in (0, 1]$.*

*Then we have:*

$$f(x, W, a) := \kappa \frac{1}{\sqrt{m}} \sum_{r=1}^m a_r \cdot \mathsf{ReLU}\Big(\langle w_r, x \rangle + \langle u(w_r), x \rangle\Big)$$

*Proof.* We have

$$f(x, W, a) = \kappa \frac{1}{\sqrt{m}} \sum_{r=1}^{m} a_r \cdot \mathsf{ReLU}\Big(\mathsf{dq}(\langle \widetilde{w}_r, x \rangle)\Big)$$

$$= \kappa \frac{1}{\sqrt{m}} \sum_{r=1}^{m} a_r \cdot \mathsf{ReLU}\Big(\sqrt{V(w)} \cdot (\langle \widetilde{w}, x \rangle + E(w) \cdot \langle x, \mathbf{1}_d \rangle)\Big)$$

$$= \kappa \frac{1}{\sqrt{m}} \sum_{r=1}^{m} a_r \cdot \mathsf{ReLU}\Big(\langle w_r, x \rangle + \langle u(w_r), x \rangle\Big)$$

where the first step follows from Definition C.5, the second step follows from Definition D.5, the last step follows from Definition D.6. $\square$

## C.3 Training

**Training.**

**Definition C.7.** *If the following conditions hold:*

- *Let $\mathcal{D} = \{(x_i, y_i)\}_{i=1}^{n} \subset \mathbb{R}^d \times \mathbb{R}$ be defined as Definition C.1.*
- *Let $W(0) \in \mathbb{R}^{d \times m}$ be initialized as Definition C.3.*
- *Let $a \in \mathbb{R}^m$ be initialized as Definition C.3.*
- *Let $f : \mathbb{R}^d \times \mathbb{R}^{d \times m} \times \mathbb{R}^m \to \mathbb{R}$ be defined as Definition C.5.*
- *For any $t \geq 0$.*

*We define:*

$$L(W(t)) := \frac{1}{2} \cdot \sum_{i=1}^{n} (f(x_i, W(t), a) - y_i)^2$$

**Definition C.8.** *If the following conditions hold:*

- *Let $\mathcal{D} = \{(x_i, y_i)\}_{i=1}^{n} \subset \mathbb{R}^d \times \mathbb{R}$ be defined as Definition C.1.*
- *Let $W(0) \in \mathbb{R}^{d \times m}$ be initialized as Definition C.3.*
- *Let $a \in \mathbb{R}^m$ be initialized as Definition C.3.*
- *Let $f : \mathbb{R}^d \times \mathbb{R}^{d \times m} \times \mathbb{R}^m \to \mathbb{R}$ be defined as Definition C.5.*
- *For any $t \geq 0$.*
- *Let $L(W(t))$ be defined as Definition C.7.*
- *Denote $\eta > 0$ as the learning rate.*
- *Let $\Delta W(t) \in \mathbb{R}^{d \times m}$ be defined as Definition F.2.*

*We update:*

$$W(t+1) := W(t) - \eta \cdot \Delta W(t)$$

**Compact Form.**

**Definition C.9.** *If the following conditions hold:*

- *Let $\mathcal{D} = \{(x_i, y_i)\}_{i=1}^{n} \subset \mathbb{R}^d \times \mathbb{R}$ be defined as Definition C.1.*

- *Let $W(0) \in \mathbb{R}^{d \times m}$ be initialized as Definition C.3.*

- *Let $a \in \mathbb{R}^m$ be initialized as Definition C.3.*

- *Let $f : \mathbb{R}^d \times \mathbb{R}^{d \times m} \times \mathbb{R}^m \to \mathbb{R}$ be defined as Definition C.5.*

- *For any $t \geq 0$.*

- *Let $L(W(t))$ be defined as Definition C.7.*

- *Let $W(t)$ be updated by Definition C.8.*

*We give the following compact form of defined variables and functions:*

- **Compact form of model function.** *We define:*
$$\mathsf{F}(t) := [f(x_1, W(t), a), f(x_2, W(t), a), \cdots, f(x_n, W(t), a)]^\top \in \mathbb{R}^n$$

- **Compact form of the input vector in the training dataset.** *We define:*
$$X := [x_1, x_2, \cdots, x_n]^\top \in \mathbb{R}^{n \times d}$$

- **Compact form of the targets in the training dataset.** *We define:*
$$y := [y_1, y_2, \cdots, y_n]^\top \in \mathbb{R}^n$$

- **Compact form of the training objective.** *We define:*
$$\mathsf{L}(t) := \frac{1}{2} \cdot \|\mathsf{F}(t) - y\|_2^2$$
*Especially, we have $\mathsf{L}(t) = L(W(t))$ by simple algebras.*

# D  Quantization

## D.1  Quantization Functions

**Definition D.1.** *For a vector $w \in \mathbb{R}^d$, we define $\mathsf{Sign}(w) \in \{-1, +1\}^d$ where its $k$-th entry for $k \in [d]$ is given by:*
$$\mathsf{Sign}_k(w) := \begin{cases} -1, & \text{if } w_k < 0 \\ +1, & \text{if } w_k \geq 0 \end{cases} \in \{-1, +1\}$$

**Definition D.2.** *For a vector $w \in \mathbb{R}^d$, we define expectation function as follows:*
$$E(w) := \frac{\langle w, \mathbf{1}_d \rangle}{d} \in \mathbb{R}$$

**Definition D.3.** *Let $E : \mathbb{R}^d \to \mathbb{R}$ be defined as Definition D.2. For a vector $w \in \mathbb{R}^d$, we define variance function as follows:*
$$V(w) := \frac{1}{d} \cdot \|w - E(w) \cdot \mathbf{1}_d\|_2^2 \in \mathbb{R}$$

**Definition D.4.** *If the following conditions hold:*

- *Let $\mathsf{Sign} : \mathbb{R}^d \to \{-1, +1\}^d$ be defined as Definition D.1.*

- *Let $E : \mathbb{R}^d \to \mathbb{R}$ be defined as Definition D.2.*

- *Let $V : \mathbb{R}^d \to \mathbb{R}$ be defined as Definition D.3.*

- *For a weight vector $w \in \mathbb{R}^d$.*

*We define the quantization function as follows:*
$$\mathsf{q}(w) := \mathsf{Sign}(\frac{w - E(w) \cdot \mathbf{1}_d}{\sqrt{V(w)}}) \in \{-1, +1\}^d$$

## D.2 Dequantization Functions

**Definition D.5.** *If the following conditions hold:*

- *Let $\mathsf{q} : \mathbb{R}^d \to \{-1, +1\}^d$ be defined as Definition D.4.*
- *Let $E : \mathbb{R}^d \to \mathbb{R}$ be defined as Definition D.2.*
- *Let $V : \mathbb{R}^d \to \mathbb{R}$ be defined as Definition D.3.*
- *For a weight vector $w \in \mathbb{R}^d$.*
- *Denote quantized vector $\widetilde{w} := \mathsf{q}(w) \in \{-1, +1\}^d$.*
- *For a vector $x \in \mathbb{R}^d$.*

*We define the dequantization function as follows:*

$$\mathsf{dq}(\langle \widetilde{w}, x \rangle) := \sqrt{V(w)} \cdot \langle \widetilde{w}, x \rangle + E(w) \cdot \langle x, \mathbf{1}_d \rangle \in \mathbb{R}$$

## D.3 Quantization Error

**Definition D.6.** *If the following conditions hold:*

- *Let $\mathsf{q} : \mathbb{R}^d \to \{-1, +1\}^d$ be defined as Definition D.4.*
- *Let $E : \mathbb{R}^d \to \mathbb{R}$ be defined as Definition D.2.*
- *Let $V : \mathbb{R}^d \to \mathbb{R}$ be defined as Definition D.3.*
- *For a weight vector $w \in \mathbb{R}^d$.*
- *Denote quantized vector $\widetilde{w} := \mathsf{q}(w) \in \{-1, +1\}^d$.*
- *For a vector $x \in \mathbb{R}^d$.*

*We define the quantization difference vector as follows:*

$$u(w) := \sqrt{V(w)}\widetilde{w} + E(w) \cdot \mathbf{1}_d - w \in \mathbb{R}^d$$

**Lemma D.7.** *If the following conditions hold:*

- *Let $D > 0$ be defined as Definition B.16.*
- *Let $\mathsf{q} : \mathbb{R}^d \to \{-1, +1\}^d$ be defined as Definition D.4.*
- *Let $E : \mathbb{R}^d \to \mathbb{R}$ be defined as Definition D.2.*
- *Let $V : \mathbb{R}^d \to \mathbb{R}$ be defined as Definition D.3.*
- *For a weight vector $w \in \mathbb{R}^d$.*
- *Denote quantized vector $\widetilde{w} := \mathsf{q}(w) \in \{-1, +1\}^d$.*
- *For a vector $x \in \mathbb{R}^d$ and $\|x\|_2 = 1$.*
- *Let $u : \mathbb{R}^d \to \mathbb{R}^d$ be defined as Definition D.6.*

*Then we have:*

$$\langle u(w), x \rangle \le O\Big(d(D + R)\Big)$$

*Proof.* We define:

$$\mathsf{Ln}(w) = \frac{w - E(w)\mathbf{1}_d}{\sqrt{V(w)}}$$

Then by simple algebras, we can show that:

$$\frac{1}{d}\|\mathsf{Ln}(w)\|_2^2 = \frac{1}{d}\left\|\frac{w - E(w)\mathbf{1}_d}{\sqrt{V(w)}}\right\|_2^2 < \frac{1}{d}\frac{\|w - E(w)\mathbf{1}_d\|_2^2}{V(w)} < 1 \tag{4}$$

Thus, we obtain:

$$\|\mathsf{Ln}(w)\|_\infty \le \|\mathsf{Ln}(w)\|_2$$
$$= (\|\mathsf{Ln}(w)\|_2^2)^{\frac{1}{2}}$$
$$< \sqrt{d}$$

where these steps follow from simple algebras and Eq. (4).

Finally, we can get that

$$|\langle u(w), x\rangle| = \sqrt{V(w)} \cdot |\langle \widetilde{w} - \mathsf{Ln}(w), x\rangle|$$
$$= O(D + R) \cdot |\langle \widetilde{w} - \mathsf{Ln}(w), x\rangle|$$
$$\le O(D + R) \cdot \|\widetilde{w} - \mathsf{Ln}(w)\|_2$$
$$= O(D + R) \cdot \Big(\sum_{k=1}^{d}(\widetilde{w}_k - \mathsf{Ln}_k(w))^2\Big)^{\frac{1}{2}}$$
$$\le O(D + R) \cdot \Big(\sum_{k=1}^{d}(\max\{\sqrt{d} - 1, 1\})^2\Big)^{\frac{1}{2}}$$
$$\le O\Big(d(D + R)\Big)$$

where the first step follows from Definition D.6, the second step follows from Part 7 of Lemma I.6, the third step follows from Cauchy-Schwarz inequality and $\|x\|_2 = 1$, the fourth step follows from the definition of $\ell_2$ norm, the fifth step follows from Definition D.1 and simple algebras, the last step follows from simple algebras. □

# E   Patterns

## E.1   ReLU Pattern

**Definition E.1.** *If the following conditions hold:*

- *For any $w \in \mathbb{R}^d$.*
- *Let $\mathcal{D} = \{(x_i, y_i)\}_{i=1}^n \subset \mathbb{R}^d \times \mathbb{R}$ be defined as Definition C.1.*
- *Let $W(0) \in \mathbb{R}^{d \times m}$ be initialized as Definition C.3.*
- *Let $\mathsf{dq} : \mathbb{R} \to \mathbb{R}$ be defined as Definition D.5.*
- *For $R > 0$.*
- *For $i \in [n]$ and $r \in [m]$.*

*We define:*

$$\mathsf{A}_{i,r} := \{\exists w \in \mathbb{R}^d : \|w - w_r(0)\|_2 \le R, \mathbf{1}_{\mathsf{dq}(\langle w_r(0), x_i\rangle)\ge 0} \ne \mathbf{1}_{\mathsf{dq}(\langle w, x_i\rangle)\ge 0}\}$$

**Definition E.2.** *Let event $\mathsf{A}_{i,r}$ for $i \in [n]$ and $r \in [m]$ be defined as Definition E.1. We define:*

$$\mathcal{S}_i := \{r \in [m] : \mathbb{I}\{\mathsf{A}_{i,r}\} = 0\}$$
$$\mathcal{S}_i^\perp := [m]/\mathcal{S}_i$$

## E.2 Sign Pattern

**Definition E.3.** *If the following conditions hold:*

- *For any $w \in \mathbb{R}^d$.*
- *Let $W(0) \in \mathbb{R}^{d \times m}$ be initialized as Definition C.3.*
- *For $R > 0$.*
- *For $k \in [d]$ and $r \in [m]$.*

*We define:*

$$\mathsf{B}_{r,k} := \{\exists w \in \mathbb{R}^d : |w_k - w_{r,k}(0)| \leq R, \mathbf{1}_{w_{r,k}(0) - E(w_r(0)) \geq 0} \neq \mathbf{1}_{w_k - E(w) \geq 0}\}$$

# F Straight-Through Estimator (STE)

## F.1 STE Functions

**Definition F.1.** *If the following conditions hold:*

- *For a input vector $x \in \mathbb{R}^d$.*
- *For a hidden-layer weights $W \in \mathbb{R}^{d \times m}$ as Definition C.2.*
- *For a output-layer weights $a \in \mathbb{R}^m$ as Definition C.2.*
- *Let $\mathsf{q} : \mathbb{R}^d \to \{-1, +1\}^d$ be defined as Definition D.4.*
- *Denote $\widetilde{w}_r = \mathsf{q}(w_r) \in \{-1, +1\}^d$.*
- *Let $\mathsf{ReLU} : \mathbb{R} \to \mathbb{R}$ be defined as Definition C.4.*

*We define:*

$$f_{\text{ste}}(x, W, a) := \kappa \frac{1}{\sqrt{m}} \sum_{r=1}^{m} a_r \cdot \mathbf{1}_{\mathsf{dq}(\langle \widetilde{w}_r, x \rangle) \geq 0} \cdot \langle w_r, x \rangle \in \mathbb{R}$$

*Then its compact form is given by*

$$\mathsf{F}_{\text{ste}}(t) := [f_{\text{ste}}(x_1, W(t), a), f_{\text{ste}}(x_2, W(t), a), \cdots, f_{\text{ste}}(x_n, W(t), a)]^\top \in \mathbb{R}^n$$

**Definition F.2.** *Let $W(0) \in \mathbb{R}^{d \times m}$ be initialized as Definition C.3. For any $t \geq 0$. We define:*

$$\Delta W(t) := \sum_{i=1}^{n} (\mathsf{F}_i(t) - y_i) \cdot \frac{\mathrm{d}\mathsf{F}_{\text{ste},i}(t)}{\mathrm{d}W(t)}$$

## F.2 Gradient Computation

**Lemma F.3.** *If the following conditions hold:*

- *For $i \in [n]$, $r \in [m]$ and integer $t \geq 0$.*
- *Let $\mathcal{D} = \{(x_i, y_i)\}_{i=1}^n \subset \mathbb{R}^d \times \mathbb{R}$ be defined as Definition C.1.*
- *Let $W(t) \in \mathbb{R}^{d \times m}$ be initialized as Definition C.3 and be updated by Definition C.8.*
- *Let $a \in \mathbb{R}^m$ be initialized as Definition C.3.*
- *Let $\mathsf{F}_{\text{ste}}(t)$ be defined as Definition F.1.*
- *Let $\mathsf{q} : \mathbb{R}^d \to \{-1, +1\}^d$ be defined as Definition D.4.*

- *Denote $\widetilde{w}_r = \mathsf{q}(w_r) \in \{-1, +1\}^d$.*

- *For $\kappa \in (0, 1]$.*

*Then we have:*

$$\frac{\mathrm{dF}_{\mathrm{ste},i}(t)}{\mathrm{d}w_r(t)} = \kappa \frac{1}{\sqrt{m}} a_r \cdot \mathbf{1}_{\mathsf{dq}(\langle \widetilde{w}_r(t), x_i \rangle) \geq 0} \cdot x_i$$

*Proof.* This proof follows from simple calculations. $\qquad\square$

# G    Neural Tangent Kernel

## G.1    Kernel Function

**Definition G.1.** *If the following conditions hold:*

- *For $i, j \in [n]$, $r \in [m]$ and integer $t \geq 0$.*

- *Let $\mathcal{D} = \{(x_i, y_i)\}_{i=1}^n \subset \mathbb{R}^d \times \mathbb{R}$ be defined as Definition C.1.*

- *Let $W(t) \in \mathbb{R}^{d \times m}$ be initialized as Definition C.3 and be updated by Definition C.8.*

- *Let $a \in \mathbb{R}^m$ be initialized as Definition C.3.*

- *Let $\mathsf{q} : \mathbb{R}^d \to \{-1, +1\}^d$ be defined as Definition D.4.*

- *Let $\mathsf{dq} : \mathbb{R} \to \mathbb{R}$ be defined as Definition D.5.*

- *Denote $\widetilde{w}_r = \mathsf{q}(w_r) \in \{-1, +1\}^d$.*

- *For $\kappa \in (0, 1]$.*

*We define the kernel function as $H(t) \in \mathbb{R}^{n \times n}$, where its $(i, j)$-th entry is given by:*

$$H_{i,j}(t) := \kappa^2 \frac{1}{m} x_i^\top x_j \cdot \sum_{r=1}^m \mathbf{1}_{\mathsf{dq}(\langle \widetilde{w}_r(t), x_i \rangle) \geq 0} \cdot \mathbf{1}_{\mathsf{dq}(\langle \widetilde{w}_r(t), x_j \rangle) \geq 0} \in \mathbb{R}$$

**Claim G.2.** *If the following conditions hold:*

- *For $i, j \in [n]$, $r \in [m]$ and integer $t \geq 0$.*

- *Let $\mathcal{D} = \{(x_i, y_i)\}_{i=1}^n \subset \mathbb{R}^d \times \mathbb{R}$ be defined as Definition C.1.*

- *Let $W(t) \in \mathbb{R}^{d \times m}$ be initialized as Definition C.3 and be updated by Definition C.8.*

- *Let $a \in \mathbb{R}^m$ be initialized as Definition C.3.*

- *Let $\mathsf{q} : \mathbb{R}^d \to \{-1, +1\}^d$ be defined as Definition D.4.*

- *Let $\mathsf{dq} : \mathbb{R} \to \mathbb{R}$ be defined as Definition D.5.*

- *Denote $\widetilde{w}_r = \mathsf{q}(w_r) \in \{-1, +1\}^d$.*

- *Let $H(t) \in \mathbb{R}^{n \times n}$ be defined as Definition G.1.*

- *For $\kappa \in (0, 1]$.*

*We first define the neural tangent network as $H^* := H(0) \in \mathbb{R}^{n \times n}$, where its $(i, j)$-th entry is given by:*

$$
\begin{aligned}
H_{i,j}^* &:= H_{i,j}(0) \\
&= \kappa^2 \frac{1}{m} x_i^\top x_j \cdot \sum_{r=1}^m \mathbf{1}_{\mathsf{dq}(\langle \widetilde{w}_r(0), x_i \rangle) \geq 0} \cdot \mathbf{1}_{\mathsf{dq}(\langle \widetilde{w}_r(0), x_j \rangle) \geq 0}
\end{aligned}
$$

$$\approx \kappa^2 x_i^\top x_j \cdot \mathop{\mathbb{E}}_{w_r \sim \mathcal{N}(0,\sigma^2 \cdot I_d)}\left[\mathbf{1}_{\mathsf{dq}(\langle \widetilde{w}_r(0), x_i\rangle)\geq 0} \cdot \mathbf{1}_{\mathsf{dq}(\langle \widetilde{w}_r(0), x_j\rangle)\geq 0}\right]$$

*Proof.* We have

$$H^*_{i,j} = H_{i,j}(0)$$

$$= \kappa^2 \frac{1}{m} x_i^\top x_j \cdot \sum_{r=1}^m \mathbf{1}_{\mathsf{dq}(\langle \widetilde{w}_r(0), x_i\rangle)\geq 0} \cdot \mathbf{1}_{\mathsf{dq}(\langle \widetilde{w}_r(0), x_j\rangle)\geq 0}$$

$$\approx \kappa^2 x_i^\top x_j \cdot \mathop{\mathbb{E}}_{w_r \sim \mathcal{N}(0,\sigma^2 \cdot I_d)}\left[\mathbf{1}_{\mathsf{dq}(\langle \widetilde{w}_r(0), x_i\rangle)\geq 0} \cdot \mathbf{1}_{\mathsf{dq}(\langle \widetilde{w}_r(0), x_j\rangle)\geq 0}\right]$$

where the first step follows from the definition of $H^*$, the second step follows from Definition G.1, the third step holds since $m \to +\infty$. $\qquad\square$

**Definition G.3.** *If the following conditions hold:*

- *For $i, j \in [n]$, $r \in [m]$ and integer $t \geq 0$.*

- *Let $\mathcal{D} = \{(x_i, y_i)\}_{i=1}^n \subset \mathbb{R}^d \times \mathbb{R}$ be defined as Definition C.1.*

- *Let $W(t) \in \mathbb{R}^{d \times m}$ be initialized as Definition C.3 and be updated by Definition C.8.*

- *Let $a \in \mathbb{R}^m$ be initialized as Definition C.3.*

- *Let $\mathsf{q} : \mathbb{R}^d \to \{-1, +1\}^d$ be defined as Definition D.4.*

- *Let $\mathsf{dq} : \mathbb{R} \to \mathbb{R}$ be defined as Definition D.5.*

- *Denote $\widetilde{w}_r = \mathsf{q}(w_r) \in \{-1, +1\}^d$.*

- *Let $\mathcal{S}_i^\perp$ be defined as Definition E.2.*

*We the pattern-changing kernel function as $H^\perp(t) \in \mathbb{R}^{n \times n}$, where its $(i, j)$-th entry is given by:*

$$H_{i,j}^\perp(t) := \kappa^2 \frac{1}{m} x_i^\top x_j \cdot \sum_{r \in \mathcal{S}_i^\perp} \mathbf{1}_{\mathsf{dq}(\langle \widetilde{w}_r(t), x_i\rangle)\geq 0} \cdot \mathbf{1}_{\mathsf{dq}(\langle \widetilde{w}_r(t), x_j\rangle)\geq 0} \in \mathbb{R}$$

## G.2 Assumption: $H^*$ is Positive Definite

**Assumption G.4.** *Let $H^* \in \mathbb{R}^{n \times n}$ be defined as Definition G.1. We assume that $H^*$ is positive definite (PD), where its minimum eigenvalue is given by:*

$$\lambda := \lambda_{\min}(H^*) > 0$$

## G.3 Kernel Convergence and PD Property

**Lemma G.5.** *If the following conditions hold:*

- *Let $D > 0$ be defined as Definition B.16.*

- *Denote $\lambda = \lambda_{\min}(H^*) > 0$ as Assumption G.4.*

- *For $i, j \in [n]$, $r \in [m]$ and integer $t \geq 0$.*

- *Let $\mathcal{D} = \{(x_i, y_i)\}_{i=1}^n \subset \mathbb{R}^d \times \mathbb{R}$ be defined as Definition C.1.*

- *Let $W(t) \in \mathbb{R}^{d \times m}$ be initialized as Definition C.3 and be updated by Definition C.8.*

- *Let $a \in \mathbb{R}^m$ be initialized as Definition C.3.*

- *Let $H(t) \in \mathbb{R}^{n \times n}$ be defined as Definition G.1.*

- *Let $H^* \in \mathbb{R}^{n \times n}$ be defined as Claim G.2.*
- $R \le O(\frac{\lambda \delta}{\kappa^2 n^2 dD})$.
- $\delta \in (0, 0.1)$.

*Then with probability at least $1 - \delta$, we have:*

- *Part 1.*

$$\|H(t) - H^*\|_F \le O\left(n^2 dR\delta^{-1}D\right)$$

- *Part 2.*

$$\lambda_{\min}(H(t)) \ge \lambda/2$$

*Proof.* **Proof of Part 1.** Let $\mathsf{A}_{i,r}$ be defined as Definition E.1, we first show that when $\langle w_r(0), x \rangle \ge R + O\left(d(D + R)\right)$

$$\begin{aligned}
\mathsf{dq}(\langle \widetilde{w}_r(0), x_i \rangle) &= \sqrt{V(w_r(0))} \cdot \langle \widetilde{w}_r(0), x_i \rangle + \langle E(w_r(0)) \cdot \mathbf{1}_d, x_i \rangle \\
&= \langle w_r(0), x_i \rangle + \langle u(w_r(0)), x_i \rangle \\
&\ge \langle w_r(0), x_i \rangle - |\langle u(w_r(0)), x_i \rangle| \\
&\ge R
\end{aligned}$$

where the first step follows from Definition D.5, the second step follows from Definition D.6. the third step follows from simple algebras, the last step follows from $\langle w_r(0), x \rangle \ge R + O\left(d(D + R)\right)$ and Lemma D.7.

Thus, for any $w \in \mathbb{R}^d$ that satisfies $\|w - w_r(0)\|_2 \le R$, we have:

$$\begin{aligned}
\mathsf{dq}(\langle \widetilde{w}, x_i \rangle) &= \sqrt{V(w)} \cdot \langle \widetilde{w}, x_i \rangle + \langle E(w) \cdot \mathbf{1}_d, x_i \rangle \\
&= \langle w, x_i \rangle + \langle u(w), x_i \rangle \\
&\ge \langle w, x_i \rangle - |\langle u(w), x_i \rangle| \\
&\ge \langle w_r(0), x_i \rangle - \|w - w_r(0)\|_2 - |\langle u(w), x_i \rangle| \\
&\ge 0
\end{aligned}$$

where the first step follows from Definition D.5, the second step follows from Definition D.6. the third step follows from simple algebras, the fourth step follows from Cauchy-Schwarz inequality and $\|x_i\| = 1$, the last step follows from $\|w - w_r(0)\|_2 \le R$, $\langle w_r(0), x \rangle \ge R + O\left(d(D + R)\right)$ and Lemma D.7.

The above situation says:

$$\begin{aligned}
\Pr\left[\mathbb{I}\{\mathsf{A}_{i,r}\} = 1\right] &\le \Pr[\langle w_r(0), x \rangle < R + O\left(d(D + R)\right)] \\
&\le \frac{4R + O\left(d(D + R)\right)}{\sqrt{2\pi}} \\
&\le O\left(dR(D + R)\right) \\
&\le O\left(dRD\right)
\end{aligned} \tag{5}$$

where the second step follows from anti-concentration of Gaussian (Fact B.3) and Fact B.4, the third step follows from simple algebras and the last step follows from plugging $R \le D$.

For $i, j \in [n]$, we have

$$\mathbb{E}[|H_{i,j}(t) - H^*_{i,j}|]$$

$$= \mathbb{E}\left[\left|\kappa^2 \frac{1}{m} x_i^\top x_j \sum_{r=1}^{m} (\mathbf{1}_{\mathsf{dq}(\langle \widetilde{w}_r(t), x_i\rangle)\geq 0} \cdot \mathbf{1}_{\mathsf{dq}(\langle \widetilde{w}_r(t), x_j\rangle)\geq 0} - \mathbf{1}_{\mathsf{dq}(\langle \widetilde{w}_r(0), x_i\rangle)\geq 0} \cdot \mathbf{1}_{\mathsf{dq}(\langle \widetilde{w}_r(0), x_j\rangle)\geq 0})\right|\right]$$

$$= \kappa^2 \frac{1}{m} \sum_{r=1}^{m} \mathbb{E}\left[\mathbf{1}_{\mathsf{dq}(\langle \widetilde{w}_r(t), x_i\rangle)\geq 0} \cdot \mathbf{1}_{\mathsf{dq}(\langle \widetilde{w}_r(t), x_j\rangle)\geq 0} - \mathbf{1}_{\mathsf{dq}(\langle \widetilde{w}_r(0), x_i\rangle)\geq 0} \cdot \mathbf{1}_{\mathsf{dq}(\langle \widetilde{w}_r(0), x_j\rangle)\geq 0}\right]$$

$$\leq \kappa^2 \frac{1}{m} \sum_{r=1}^{m} \mathbb{E}\left[\mathbb{I}\{A_{i,r} \cup A_{j,r}\}\right]$$

$$\leq O\left(\kappa^2 dRD\right) \tag{6}$$

where the first step follows from Definition G.1 and Claim G.2, the second and third step follows from simple algebras, the last step follows from Eq. (5).

Then we have:

$$\mathbb{E}[\sum_{i=1}^{n}\sum_{j=1}^{n} |H_{i,j}(t) - H_{i,j}^*|] = \sum_{i=1}^{n}\sum_{j=1}^{n} \mathbb{E}[|H_{i,j}(t) - H_{i,j}^*|]$$

$$\leq O\left(\kappa^2 n^2 dRD\right)$$

where the first step follows from simple algebras, the second step follows from Eq. (6).

Hence, by Markov's inequality (Lemma B.15), with probability at least $1 - \delta$, we have:

$$\sum_{i=1}^{n}\sum_{j=1}^{n} |H_{i,j}(t) - H_{i,j}^*| \leq \frac{\mathbb{E}[\sum_{i=1}^{n}\sum_{j=1}^{n} |H_{i,j}(t) - H_{i,j}^*|]}{\delta}$$

$$\leq O\left(\kappa^2 n^2 dR\delta^{-1}(D+R)\right)$$

We obtain:

$$\|H(t) - H^*\|_F \leq \|H(t) - H^*\|_1$$

$$= \sum_{i=1}^{n}\sum_{j=1}^{n} |H_{i,j}(t) - H_{i,j}^*|$$

$$\leq O\left(\kappa^2 n^2 dR\delta^{-1}D\right)$$

Now following Fact B.5, we have:

$$\lambda_{\min}(H(t)) \geq \lambda_{\min}(H^*) - \|H(t) - H^*\|_F$$

$$\geq \lambda - O\left(\kappa^2 n^2 dR\delta^{-1}D\right)$$

$$\geq \lambda/2$$

where the last step follows from choosing $R \leq O(\frac{\lambda\delta}{\kappa^2 n^2 dD})$. $\qquad\square$

# H  Training Dynamic

## H.1  Decompose Loss

**Definition H.1.** *Let $W(0) \in \mathbb{R}^{d \times m}$ be initialized as Definition C.3. For any $t \geq 0$. Let $u : \mathbb{R}^d \to \mathbb{R}^d$ be defined as Definition D.6. For $r \in [m]$. We define:*

$$\mathsf{u}_r(t) := u(w_r(t))$$

*Then the $\mathsf{F}_i(t), \forall i \in [n]$ can be given by:*

$$\mathsf{F}_i(t) = \kappa \frac{1}{\sqrt{m}} \sum_{r=1}^{m} a_r \cdot \mathbf{1}_{\mathsf{dq}(\langle \widetilde{w}_r(t), x_i\rangle)\geq 0} \cdot \left(\langle w_r(t), x_i\rangle + \langle \mathsf{u}_r(t), x_i\rangle\right)$$

**Claim H.2.** *If the following conditions hold:*

- *For $i, j \in [n]$, $r \in [m]$ and integer $t \geq 0$.*
- *Let $\mathsf{L}(t)$ be defined as Definition C.9.*
- *Let $\mathsf{F}(t) \in \mathbb{R}^n$ be defined as Definition C.9.*
- *Let $\mathcal{D} = \{(x_i, y_i)\}_{i=1}^n \subset \mathbb{R}^d \times \mathbb{R}$ be defined as Definition C.1.*
- *Let $W(t) \in \mathbb{R}^{d \times m}$ be initialized as Definition C.3 and be updated by Definition C.8.*
- *Let $a \in \mathbb{R}^m$ be initialized as Definition C.3.*
- *Let $\mathsf{dq} : \mathbb{R} \to \mathbb{R}$ be defined as Definition D.5.*
- *Denote $\widetilde{w}_r = \mathsf{q}(w_r) \in \{-1, +1\}^d$.*
- *Let $\mathcal{S}_i, \mathcal{S}_i^\perp$ be defined as Definition E.2.*
- *Let $\mathsf{u}_r(t)$ be defined as Definition H.1.*
- *Define*

$$C_1 := -\kappa \frac{1}{\sqrt{m}} \sum_{i=1}^n \sum_{r \in \mathcal{S}_i} a_r \big(\mathbf{1}_{\mathsf{dq}(\langle \widetilde{w}_r(t), x_i \rangle) \geq 0} \langle w_r(t), x_i \rangle$$
$$- \mathbf{1}_{\mathsf{dq}(\langle \widetilde{w}_r(t+1), x_i \rangle) \geq 0} \langle w_r(t+1), x_i \rangle\big) \cdot (\mathsf{F}_i(t) - y_i)$$

- *Define*

$$C_2 := -\kappa \frac{1}{\sqrt{m}} \sum_{i=1}^n \sum_{r \in \mathcal{S}_i^\perp} a_r \Big(\mathbf{1}_{\mathsf{dq}(\langle \widetilde{w}_r(t), x_i \rangle) \geq 0} \langle w_r(t), x_i \rangle$$
$$- \mathbf{1}_{\mathsf{dq}(\langle \widetilde{w}_r(t+1), x_i \rangle) \geq 0} \langle w_r(t+1), x_i \rangle\Big) \cdot (\mathsf{F}_i(t) - y_i)$$

- *Define*

$$C_3 := -\kappa \frac{1}{\sqrt{m}} \sum_{i=1}^n \sum_{r=1}^m a_r \Big(\mathbf{1}_{\mathsf{dq}(\langle \widetilde{w}_r(t), x_i \rangle) \geq 0} \langle \mathsf{u}_r(t), x_i \rangle$$
$$- \mathbf{1}_{\mathsf{dq}(\langle \widetilde{w}_r(t+1), x_i \rangle) \geq 0} \langle \mathsf{u}_r(t+1), x_i \rangle\Big) \cdot (\mathsf{F}_i(t) - y_i)$$

- *Define*

$$C_4 := \frac{1}{2} \|\mathsf{F}(t) - \mathsf{F}(t+1)\|_2^2$$

- *For $\kappa \in (0, 1]$.*

*Then we have:*

$$\mathsf{L}(t+1) = L(t) + C_1 + C_2 + C_3 + C_4$$

*Proof.* We have

$$\mathsf{L}(t+1) = \frac{1}{2} \cdot \|\mathsf{F}(t+1) - y\|_2^2$$
$$= \frac{1}{2} \cdot \|(\mathsf{F}(t) - y) - (\mathsf{F}(t) - \mathsf{F}(t+1))\|_2^2$$
$$= \frac{1}{2} \cdot (\|\mathsf{F}(t) - y\|_2^2 - 2\langle \mathsf{F}(t) - y, \mathsf{F}(t) - \mathsf{F}(t+1)\rangle + \|\mathsf{F}(t) - \mathsf{F}(t+1)\|_2^2)$$

$$= \mathsf{L}(t) - \langle \mathsf{F}(t) - y, \mathsf{F}(t) - \mathsf{F}(t+1) \rangle + \frac{1}{2}\|\mathsf{F}(t) - \mathsf{F}(t+1)\|_2^2$$

these steps follow from simple algebras and Definition C.9.

Then for $i \in [n]$

$$\mathsf{F}_i(t) - \mathsf{F}_i(t+1)$$

$$= \kappa \frac{1}{\sqrt{m}} \sum_{r=1}^{m} a_r \cdot \mathbf{1}_{\mathsf{dq}(\langle \widetilde{w}_r(t), x_i \rangle) \geq 0} \cdot \Big( \langle w_r(t), x_i \rangle + \langle \mathsf{u}_r(t), x_i \rangle \Big)$$

$$- \kappa \frac{1}{\sqrt{m}} \sum_{r=1}^{m} a_r \cdot \mathbf{1}_{\mathsf{dq}(\langle \widetilde{w}_r(t+1), x_i \rangle) \geq 0} \cdot \Big( \langle w_r(t+1), x_i \rangle + \langle \mathsf{u}_r(t+1), x_i \rangle \Big)$$

$$= \kappa \frac{1}{\sqrt{m}} \sum_{r=1}^{m} a_r \cdot \bigg( \mathbf{1}_{\mathsf{dq}(\langle \widetilde{w}_r(t), x_i \rangle) \geq 0} \cdot \Big( \langle w_r(t), x_i \rangle + \langle \mathsf{u}_r(t), x_i \rangle \Big)$$

$$- \mathbf{1}_{\mathsf{dq}(\langle \widetilde{w}_r(t+1), x_i \rangle) \geq 0} \cdot \Big( \langle w_r(t+1), x_i \rangle + \langle \mathsf{u}_r(t+1), x_i \rangle \Big) \bigg)$$

$$= M_{1,i} + M_{2,i} + M_{3,i}$$

where these steps follows from simple algebras and defining:

$$M_{1,i} := \kappa \frac{1}{\sqrt{m}} \sum_{r \in \mathcal{S}_i} a_r \Big( \mathbf{1}_{\mathsf{dq}(\langle \widetilde{w}_r(t), x_i \rangle) \geq 0} \cdot \langle w_r(t), x_i \rangle - \mathbf{1}_{\mathsf{dq}(\langle \widetilde{w}_r(t+1), x_i \rangle) \geq 0} \cdot \langle w_r(t+1), x_i \rangle \Big)$$

$$M_{2,i} := \kappa \frac{1}{\sqrt{m}} \sum_{r \in \mathcal{S}_i^{\perp}} a_r \Big( \mathbf{1}_{\mathsf{dq}(\langle \widetilde{w}_r(t), x_i \rangle) \geq 0} \cdot \langle w_r(t), x_i \rangle - \mathbf{1}_{\mathsf{dq}(\langle \widetilde{w}_r(t+1), x_i \rangle) \geq 0} \cdot \langle w_r(t+1), x_i \rangle \Big)$$

$$M_{3,i} := \kappa \frac{1}{\sqrt{m}} \sum_{r=1}^{m} a_r \Big( \mathbf{1}_{\mathsf{dq}(\langle \widetilde{w}_r(t), x_i \rangle) \geq 0} \cdot \langle \mathsf{u}_r(t), x_i \rangle - \mathbf{1}_{\mathsf{dq}(\langle \widetilde{w}_r(t+1), x_i \rangle) \geq 0} \cdot \langle \mathsf{u}_r(t+1), x_i \rangle \Big)$$

Thus, by the definitions in Lemma conditions, we can show that

$$\mathsf{L}(t+1) = \mathsf{L}(t) + C_1 + C_2 + C_3 + C_4$$

$\square$

## H.2 Bounding $C_1$

**Lemma H.3.** *If the following conditions hold:*

- *Let $D > 0$ be defined as Definition B.16.*
- *For $i, j \in [n]$, $r \in [m]$ and integer $t \geq 0$.*
- *Let $H(t) \in \mathbb{R}^{n \times n}$ be defined as Definition G.1.*
- *Let $H^{\perp}(t) \in \mathbb{R}^{n \times n}$ be defined as Definition G.3.*
- *Let $H^* \in \mathbb{R}^{n \times n}$ be defined as Claim G.2. Assume $\lambda_{\min}(H^*) > 0$ as Assumption G.4.*
- *Let $\mathsf{L}(t)$ be defined as Definition C.9.*
- *Let $\mathsf{F}(t) \in \mathbb{R}^n$ be defined as Definition C.9.*
- *Let $\mathcal{D} = \{(x_i, y_i)\}_{i=1}^n \subset \mathbb{R}^d \times \mathbb{R}$ be defined as Definition C.1.*
- *Let $W(t) \in \mathbb{R}^{d \times m}$ be initialized as Definition C.3 and be updated by Definition C.8.*
- *Let $a \in \mathbb{R}^m$ be initialized as Definition C.3.*

- *Let* $\mathsf{dq} : \mathbb{R} \to \mathbb{R}$ *be defined as Definition D.5.*

- *Denote* $\widetilde{w}_r = \mathsf{q}(w_r) \in \{-1, +1\}^d$.

- *Let* $\mathcal{S}_i, \mathcal{S}_i^\perp$ *be defined as Definition E.2.*

- *Let* $\mathsf{u}_r(t)$ *be defined as Definition H.1.*

- $\delta \in (0, 0.1)$.

- *Define*

$$
\begin{aligned}
C_1 := & -\kappa \frac{1}{\sqrt{m}} \sum_{i=1}^n \sum_{r \in \mathcal{S}_i} a_r (\mathbf{1}_{\mathsf{dq}(\langle \widetilde{w}_r(t), x_i \rangle) \geq 0} \langle w_r(t), x_i \rangle \\
& - \mathbf{1}_{\mathsf{dq}(\langle \widetilde{w}_r(t+1), x_i \rangle) \geq 0} \langle w_r(t+1), x_i \rangle) \cdot (\mathsf{F}_i(t) - y_i)
\end{aligned}
$$

- *For* $\kappa \in (0, 1]$.

*Then with probability at least* $1 - \delta$*, we have:*

$$
C_1 \leq \Big( -\eta\kappa\lambda + O(\eta\kappa \frac{n^2 dRD}{\delta}) \Big) \cdot \mathsf{L}(t)
$$

*Proof.* We have:

$$
\begin{aligned}
C_1 = & -\kappa \frac{1}{\sqrt{m}} \sum_{i=1}^n \sum_{r \in \mathcal{S}_i} a_r (\mathbf{1}_{\mathsf{dq}(\langle \widetilde{w}_r(t), x_i \rangle) \geq 0} \langle w_r(t), x_i \rangle \\
& - \mathbf{1}_{\mathsf{dq}(\langle \widetilde{w}_r(t+1), x_i \rangle) \geq 0} \langle w_r(t+1), x_i \rangle) \cdot (\mathsf{F}_i(t) - y_i) \\
= & -\kappa \frac{1}{\sqrt{m}} \sum_{i=1}^n \sum_{r \in \mathcal{S}_i} a_r (\langle w_r(t), x_i \rangle - \langle w_r(t+1), x_i \rangle) \cdot (\mathsf{F}_i(t) - y_i) \\
= & -\kappa^2 \eta \frac{1}{m} \sum_{i=1}^n \sum_{r \in \mathcal{S}_i} (\mathsf{F}_i(t) - y_i) \cdot (\sum_{j=1}^n x_i^\top x_j \cdot \mathbf{1}_{\mathsf{dq}(\langle \widetilde{w}_r(t), x_i \rangle) \geq 0} \cdot \mathbf{1}_{\mathsf{dq}(\langle \widetilde{w}_r(t), x_j \rangle) \geq 0} \cdot (\mathsf{F}_j(t) - y_j)) \\
= & -\eta(\mathsf{F}(t) - y)^\top \cdot (H(t) - H^\perp(t)) \cdot (\mathsf{F}(t) - y) \\
= & -\eta(\mathsf{F}(t) - y)^\top \cdot H(t) \cdot (\mathsf{F}(t) - y) + \eta(\mathsf{F}(t) - y)^\top \cdot H^\perp(t) \cdot (\mathsf{F}(t) - y) \\
\leq & -\eta\lambda/2 \cdot \|\mathsf{F}(t) - y\|_2^2 + \eta\|H^\perp(t)\|_F \cdot \|\mathsf{F}(t) - y\|_2 \\
= & (-\eta\lambda + \|H^\perp(t)\|_F) \cdot \mathsf{L}(t)
\end{aligned}
$$

where the first step follows from definition of $C_1$, the second step follows from the definition of $\mathcal{S}_i$ (Definition E.2), the third step follows from Definition C.8 and Definition F.2, the fourth step follows from Definition G.1, Definition G.3 and simple algebras, the fifth step follows from simple algebras, the sixth step follows from Lemma G.5 and simple algebras, the last step follows from Definition C.9.

Besides, we have

$$
\begin{aligned}
|H_{i,j}^\perp| &= |\frac{1}{m} x_i^\top x_j \cdot \sum_{r \in \mathcal{S}_i^\perp} \mathbf{1}_{\mathsf{dq}(\langle \widetilde{w}_r(t), x_i \rangle) \geq 0} \cdot \mathbf{1}_{\mathsf{dq}(\langle \widetilde{w}_r(t), x_j \rangle) \geq 0}| \\
&\leq |\frac{1}{m} x_i^\top x_j \cdot |\mathcal{S}_i^\perp|| \\
&\leq \frac{1}{m} |\mathcal{S}_i^\perp| \tag{7}
\end{aligned}
$$

where the first step follows from Definition G.3, the second step follows from simple algebras, the third step follows from $\|x\|_i = 1$.

We give that

$$\mathbb{E}[\sum_{i=1}^{n} |\mathcal{S}_i^{\perp}|] = \sum_{i=1}^{n} \sum_{r=1}^{m} \Pr[\mathbb{I}\{A_{i,r}\} = 1]$$
$$\leq O(mndRD)$$

where the first step follows from simple algebras, the second step follows from Eq. (5).

Hence, by Markov's inequality (Lemma B.15), we have

$$\sum_{i=1}^{n} |\mathcal{S}_i^{\perp}| \leq O(\frac{mndRD}{\delta}) \tag{8}$$

Thus,

$$\|H^{\perp}\|_F \leq \sum_{i=1}^{n} \sum_{j=1}^{n} |H_{i,j}^{\perp}|$$
$$\leq \frac{1}{m} \sum_{i=1}^{n} \sum_{j=1}^{n} |\mathcal{S}_i^{\perp}|$$
$$\leq O(\frac{n^2 dRD}{\delta})$$

where the first step follows from simple algebras, the second step follows from Eq. (7), the last step follows from simple algebras and Eq. (8).

Finally, we conclude all the results, we have:

$$C_1 \leq \left( -\eta\lambda + O(\eta\frac{n^2 dRD}{\delta}) \right) \cdot \mathsf{L}(t)$$

$\square$

## H.3 Bounding $C_2$

**Lemma H.4.** *If the following conditions hold:*

- *Let $D > 0$ be defined as Definition B.16.*
- *For $i, j \in [n]$, $r \in [m]$ and integer $t \geq 0$.*
- *Let $H(t) \in \mathbb{R}^{n \times n}$ be defined as Definition G.1.*
- *Let $H^{\perp}(t) \in \mathbb{R}^{n \times n}$ be defined as Definition G.3.*
- *Let $H^* \in \mathbb{R}^{n \times n}$ be defined as Claim G.2. Assume $\lambda_{\min}(H^*) > 0$ as Assumption G.4.*
- *Let $\mathsf{L}(t)$ be defined as Definition C.9.*
- *Let $\mathsf{F}(t) \in \mathbb{R}^n$ be defined as Definition C.9.*
- *Let $\mathcal{D} = \{(x_i, y_i)\}_{i=1}^{n} \subset \mathbb{R}^d \times \mathbb{R}$ be defined as Definition C.1.*
- *Let $W(t) \in \mathbb{R}^{d \times m}$ be initialized as Definition C.3 and be updated by Definition C.8.*
- *Let $a \in \mathbb{R}^m$ be initialized as Definition C.3.*
- *Let $\mathsf{dq} : \mathbb{R} \to \mathbb{R}$ be defined as Definition D.5.*
- *Denote $\widetilde{w}_r = \mathsf{q}(w_r) \in \{-1, +1\}^d$.*
- *Let $\mathcal{S}_i, \mathcal{S}_i^{\perp}$ be defined as Definition E.2.*

- *Let $u_r(t)$ be defined as Definition H.1.*

- $\delta \in (0, 0.1)$.

- *Define*

$$C_2 := -\kappa \frac{1}{\sqrt{m}} \sum_{i=1}^{n} \sum_{r \in \mathcal{S}_i^{\perp}} a_r \left( \mathbf{1}_{\mathsf{dq}(\langle \widetilde{w}_r(t), x_i \rangle) \geq 0} \langle w_r(t), x_i \rangle \right.$$

$$\left. - \mathbf{1}_{\mathsf{dq}(\langle \widetilde{w}_r(t+1), x_i \rangle) \geq 0} \langle w_r(t+1), x_i \rangle \right) \cdot (\mathsf{F}_i(t) - y_i)$$

- $\kappa \in (0, 1]$.

*Then with probability at least $1 - \delta$, we have:*

$$|C_2| \leq O(\eta \kappa \frac{n^{1.5} dRD}{\delta}) \cdot \mathsf{L}(t)$$

*Proof.* We have:

$$|C_2| = |\kappa \frac{1}{\sqrt{m}} \sum_{i=1}^{n} \sum_{r \in \mathcal{S}_i^{\perp}} a_r \left( \mathbf{1}_{\mathsf{dq}(\langle \widetilde{w}_r(t), x_i \rangle) \geq 0} \langle w_r(t), x_i \rangle \right.$$

$$\left. - \mathbf{1}_{\mathsf{dq}(\langle \widetilde{w}_r(t+1), x_i \rangle) \geq 0} \langle w_r(t+1), x_i \rangle \right) \cdot (\mathsf{F}_i(t) - y_i)|$$

$$\leq |\kappa \frac{1}{\sqrt{m}} \sum_{i=1}^{n} |\mathcal{S}_{i^{\perp}}| \cdot |\langle w_r(t), x_i \rangle - \langle w_r(t+1), x_i \rangle| \cdot (\mathsf{F}_i(t) - y_i)|$$

$$\leq |\kappa \frac{1}{\sqrt{m}} \sum_{i=1}^{n} |\mathcal{S}_{i^{\perp}}| \cdot \|\eta \Delta w_r(t)\|_2 \cdot (\mathsf{F}_i(t) - y_i)|$$

$$\leq \kappa \frac{1}{\sqrt{m}} \sum_{i=1}^{n} |\mathcal{S}_{i^{\perp}}| \cdot \|\eta \Delta w_r(t)\|_2 \|\mathsf{F}(t) - y\|_2$$

$$\leq \eta \kappa \frac{\sqrt{n}}{m} \sum_{i=1}^{n} |\mathcal{S}_{i^{\perp}}| \cdot \|\mathsf{F}(t) - y\|_2^2$$

$$\leq O(\eta \kappa \frac{n^{1.5} dRD}{\delta}) \cdot \mathsf{L}(t)$$

where the first step follows from the definition of $C_2$, the second step follows from Fact B.2 and Definition E.2 ($S_i^{\perp}$), the third step follows from simple algebras and Definition C.8, the fourth step follows from simple algebras, the fifth step follows from Lemma I.4, last step follows from Eq. (8) and Definition C.9. $\qquad\square$

## H.4 Bounding $C_3$

**Lemma H.5.** *If the following conditions hold:*

- *Let $D > 0$ be defined as Definition B.16.*

- *For $i, j \in [n]$, $r \in [m]$ and integer $t \geq 0$.*

- *Let $H(t) \in \mathbb{R}^{n \times n}$ be defined as Definition G.1.*

- *Let $H^{\perp}(t) \in \mathbb{R}^{n \times n}$ be defined as Definition G.3.*

- *Let $H^* \in \mathbb{R}^{n \times n}$ be defined as Claim G.2. Assume $\lambda_{\min}(H^*) > 0$ as Assumption G.4.*

- *Let $\mathsf{L}(t)$ be defined as Definition C.9.*

- Let $\mathsf{F}(t) \in \mathbb{R}^n$ be defined as Definition C.9.

- Let $\mathcal{D} = \{(x_i, y_i)\}_{i=1}^n \subset \mathbb{R}^d \times \mathbb{R}$ be defined as Definition C.1.

- Let $W(t) \in \mathbb{R}^{d \times m}$ be initialized as Definition C.3 and be updated by Definition C.8.

- Let $a \in \mathbb{R}^m$ be initialized as Definition C.3.

- Let $\mathsf{dq} : \mathbb{R} \to \mathbb{R}$ be defined as Definition D.5.

- Denote $\widetilde{w}_r = \mathsf{q}(w_r) \in \{-1, +1\}^d$.

- Let $\mathcal{S}_i, \mathcal{S}_i^\perp$ be defined as Definition E.2.

- Let $\mathsf{u}_r(t)$ be defined as Definition H.1.

- $\delta \in (0, 0.1)$.

- For an error $\epsilon > 0$ and $\|\mathsf{F}(t) - y\|_2 \geq c \cdot \epsilon$ for a sufficient small constant $c > 0$.

- Define

$$
C_3 := -\kappa \frac{1}{\sqrt{m}} \sum_{i=1}^n \sum_{r=1}^m a_r \Big( \mathbf{1}_{\mathsf{dq}(\langle \widetilde{w}_r(t), x_i \rangle) \geq 0} \langle \mathsf{u}_r(t), x_i \rangle \\
- \mathbf{1}_{\mathsf{dq}(\langle \widetilde{w}_r(t+1), x_i \rangle) \geq 0} \langle \mathsf{u}_r(t+1), x_i \rangle \Big) \cdot (\mathsf{F}_i(t) - y_i)
$$

- $\kappa \in (0, 1]$.

*Then with probability at least $1 - \delta$, we have:*

$$
C_3 \leq O\Big( \eta \kappa \frac{R^2 n^{1.5} \sqrt{d}}{\delta \epsilon \sqrt{m}} D \Big) \cdot \mathsf{L}(t)
$$

*Proof.* We have:

$$
\begin{aligned}
& |\mathsf{u}_{r,k}(t) - \mathsf{u}_{r,k}(t+1)| \\
&= |\sqrt{V(w_r(t))} \cdot \widetilde{w}_{r,k}(t) + E(w_r(t)) - w_{r,k}(t) \\
& \quad - \sqrt{V(w_r(t+1))} \cdot \widetilde{w}_{r,k}(t+1) - E(w_r(t+1)) + w_{r,k}(t+1)| \\
&\leq |\widetilde{w}_{r,k}(t) \sqrt{V(w_r(t))} - \widetilde{w}_{r,k}(t+1) \sqrt{V(w_r(t+1))}| \\
& \quad + |\eta E(\Delta w_r(t))| + |\eta \Delta w_{r,k}(t)| \\
&\leq \Big| \widetilde{w}_{r,k}(t+1)(\sqrt{V(w_r(t))} - \sqrt{V(w_r(t+1))}) \Big| \\
& \quad + \Big| \sqrt{V(w_r(t))}(\widetilde{w}_{r,k}(t) - \widetilde{w}_{r,k}(t+1)) \Big| + |\eta E(\Delta w_r(t))| + |\eta \Delta w_{r,k}(t)| \\
&= Q_{1,r,k} + Q_{2,r,k} + Q_{3,r,k} + Q_{4,r,k}
\end{aligned} \tag{9}
$$

where the first step follows from Definition H.1, the second step follows from triangle inequality and Definition C.8, the third step follows from simple algebras, the last step follows from defining:

$$
\begin{aligned}
Q_{1,r,k} &:= \Big| \widetilde{w}_{r,k}(t+1)(\sqrt{V(w_r(t))} - \sqrt{V(w_r(t+1))}) \Big| \\
Q_{2,r,k} &:= \Big| \sqrt{V(w_r(t))}(\widetilde{w}_{r,k}(t) - \widetilde{w}_{r,k}(t+1)) \Big| \\
Q_{3,r,k} &:= |\eta E(\Delta w_r(t))| \\
Q_{4,r,k} &:= |\eta \Delta w_{r,k}(t)|
\end{aligned}
$$

**Bounding $Q_{1,r,k}$.**

We have:

$$Q_{1,r,k} = \left| \widetilde{w}_{r,k}(t+1)(\sqrt{V(w_r(t))} - \sqrt{V(w_r(t+1))}) \right|$$

$$= \left| (\sqrt{V(w_r(t))} - \sqrt{V(w_r(t+1))}) \right|$$

$$\leq \| w_r(t) - E(w_r(t))\mathbf{1}_d - w_r(t+1) + E(w_r(t+1))\mathbf{1}_d \|_2$$

$$\leq \| \eta \Delta w_r(t) \|_2 + \sqrt{d} \cdot |\eta E(\Delta w_r(t))|$$

$$\leq \eta \frac{(1+\sqrt{d})\sqrt{n}}{\sqrt{m}} \| \mathsf{F}(t) - y \|_2$$

where the first step follows from the definition of $Q_{1,r,k}$, the second step follows from $\widetilde{w}_{r,k}(t+1) \in \{-1,+1\}$, the third step follows from Definition D.3 and reverse triangle inequality, the fourth step follows from $\|\mathbf{1}_d\|_2 = \sqrt{d}$ and Definition C.8, the last step follows from Lemma I.4.

**Bounding $Q_{2,r,k}$.**

We have:

$$Q_{2,r,k} = \left| \sqrt{V(w_r(t))}(\widetilde{w}_{r,k}(t) - \widetilde{w}_{r,k}(t+1)) \right|$$

$$= |\sqrt{V(w_r(t))}| \cdot |\widetilde{w}_{r,k}(t) - \widetilde{w}_{r,k}(t+1)|$$

$$\leq \| w_r(t) - E(w_r(t))\mathbf{1}_d \| \cdot |\widetilde{w}_{r,k}(t) - \widetilde{w}_{r,k}(t+1)|$$

$$\leq O(\sqrt{d}D + R) \cdot |\widetilde{w}_{r,k}(t) - \widetilde{w}_{r,k}(t+1)| \tag{10}$$

where the first step follows from the definition of $Q_{2,r,k}$, the second step follows from simple algebras, the third step follows from Definition D.3, the last step follows from Part 2 of Lemma I.6.

At the same time, we can show that

$$\mathbb{E}[|\widetilde{w}_{r,k}(t) - \widetilde{w}_{r,k}(t+1)|]$$

$$\leq 2(1 - \Pr[\mathbb{I}\{\mathsf{B}_{r,k}\} = 0 \cap \mathbb{I}\{|w_{r,k}(t) - E(w_r(t))| \geq |\eta \Delta w_{r,k}(t) - \eta E(\Delta w_r(t))|\}])$$

$$\leq 2(1 - \Pr[z \geq 2R + 2\eta \frac{\sqrt{n}}{\sqrt{m}} \| \mathsf{F}(t) - y \|_2])$$

$$= 2\Pr[z \leq 2R + 2\eta \frac{\sqrt{n}}{\sqrt{m}} \| \mathsf{F}(t) - y \|_2]$$

$$\leq O(\eta \frac{\sqrt{n}}{\sqrt{m}}) \| \mathsf{F}(t) - y \|_2 + O(1)R$$

$$\leq O(\eta \frac{R\sqrt{n}}{\epsilon \sqrt{m}}) \| \mathsf{F}(t) - y \|_2$$

where the first step follows from Definition E.3 and simple algebras, the second step follows from defining:

$$z := w_{r,k}(0) - E(w_r(0))$$

$$= \frac{d-1}{d} w_{r,k} - \frac{1}{d} \sum_{k' \in [d]/\{k\}} w_{r,k'}(0)$$

$$\sim \mathcal{N}\left(0, \sigma^2 \sqrt{\frac{d-1}{d}} \cdot I_d\right)$$

and the last steps follow from the anti-concentration of the Gaussian variable (Fact B.3) and $\| \mathsf{F}(t) - y \|_2 \geq \epsilon$ by Lemma condition.

Following Markov's inequality, we get:

$$|\widetilde{w}_{r,k}(t) - \widetilde{w}_{r,k}(t+1)| \leq O(\eta \frac{R\sqrt{n}}{\delta \epsilon \sqrt{m}}) \| \mathsf{F}(t) - y \|_2 \tag{11}$$

Hence,

$$Q_{2,r,k} \leq O\left(\eta \frac{R^2\sqrt{nd}}{\delta\epsilon\sqrt{m}} D\right) \|\mathsf{F}(t) - y\|_2$$

where this step follows from Eq. (11) and Eq. (10).

**Bounding $Q_{3,r,k}$ and $Q_{4,r,k}$.**

We can show that $Q_{3,r,k} \leq \eta \frac{\sqrt{n}}{\sqrt{m}} \cdot \|\mathsf{F}(t) - y\|_2$ and $Q_{4,r,k} \leq \eta \frac{\sqrt{n}}{\sqrt{m}} \cdot \|\mathsf{F}(t) - y\|_2$ by following Lemma I.4.

**Combination.** We have:

$$\mathbb{E}[C_3] = 0$$

where this step follows from the symmetry of $a$.

Also

$$
\begin{aligned}
&\left(\mathbf{1}_{\mathsf{dq}(\langle\widetilde{w}_r(t),x_i\rangle)\geq 0}\langle\mathsf{u}_r(t), x_i\rangle - \mathbf{1}_{\mathsf{dq}(\langle\widetilde{w}_r(t+1),x_i\rangle)\geq 0}\langle\mathsf{u}_r(t+1), x_i\rangle\right) \\
&\leq |\langle\mathsf{u}_r(t), x_i\rangle - \langle\mathsf{u}_r(t+1), x_i\rangle| \\
&= Q_{1,r,k} + Q_{2,r,k} + Q_{3,r,k} + Q_{4,r,k} \\
&\leq O\left(\eta\frac{R^2\sqrt{nd}}{\delta\epsilon\sqrt{m}}D\right)\|\mathsf{F}(t) - y\|_2
\end{aligned}
\tag{12}
$$

where the first step follows from ReLU is a 1-Lipschitz function (Fact B.2), the last step follows from simple algebras and the combination of these terms.

By Hoeffding's inequality (Lemma B.8), with a probability at least $1 - \delta$, we have:

$$
\begin{aligned}
|C_3| &\leq O\left(\eta\kappa\frac{R^2 n^{1.5}\sqrt{d}}{\delta\epsilon \cdot m}\sqrt{m}D\right)\|\mathsf{F}(t) - y\|_2^2 \\
&\leq O\left(\eta\kappa\frac{R^2 n^{1.5}\sqrt{d}}{\delta\epsilon\sqrt{m}}D\right) \cdot \mathsf{L}(t)
\end{aligned}
$$

$\square$

## H.5 Bounding $C_4$

**Lemma H.6.** *If the following conditions hold:*

- *Let $D > 0$ be defined as Definition B.16.*
- *For $i, j \in [n]$, $r \in [m]$ and integer $t \geq 0$.*
- *Let $H(t) \in \mathbb{R}^{n\times n}$ be defined as Definition G.1.*
- *Let $H^\perp(t) \in \mathbb{R}^{n\times n}$ be defined as Definition G.3.*
- *Let $H^* \in \mathbb{R}^{n\times n}$ be defined as Claim G.2. Assume $\lambda_{\min}(H^*) > 0$ as Assumption G.4.*
- *Let $\mathsf{L}(t)$ be defined as Definition C.9.*
- *Let $\mathsf{F}(t) \in \mathbb{R}^n$ be defined as Definition C.9.*
- *Let $\mathcal{D} = \{(x_i, y_i)\}_{i=1}^n \subset \mathbb{R}^d \times \mathbb{R}$ be defined as Definition C.1.*
- *Let $W(t) \in \mathbb{R}^{d\times m}$ be initialized as Definition C.3 and be updated by Definition C.8.*
- *Let $a \in \mathbb{R}^m$ be initialized as Definition C.3.*
- *Let $\mathsf{dq} : \mathbb{R} \to \mathbb{R}$ be defined as Definition D.5.*

- *Denote $\widetilde{w}_r = \mathsf{q}(w_r) \in \{-1, +1\}^d$.*
- *Let $\mathcal{S}_i, \mathcal{S}_i^{\perp}$ be defined as Definition E.2.*
- *Let $\mathsf{u}_r(t)$ be defined as Definition H.1.*
- *$\delta \in (0, 0.1)$.*
- *For an error $\epsilon > 0$ and $\|\mathsf{F}(t) - y\|_2 \geq c \cdot \epsilon$ for a sufficient small constant $c > 0$.*
- *Define*

$$C_4 := \frac{1}{2}\|\mathsf{F}(t) - \mathsf{F}(t+1)\|_2^2$$

*Then with probability at least $1 - \delta$, we have:*

$$|C_4| \leq O\left(\eta^2 \kappa^2 \frac{R^4 n^2 d}{\delta^2 \epsilon^2 m} D^2\right) \mathsf{L}(t)$$

*Proof.* We have:

$$
\begin{aligned}
&|\mathbf{1}_{\mathsf{dq}(\langle \widetilde{w}_r(t), x_i\rangle) \geq 0}(\langle w_r(t), x_i\rangle + \langle \mathsf{u}_r(t), x_i\rangle) \\
&\quad - \mathbf{1}_{\mathsf{dq}(\langle \widetilde{w}_r(t+1), x_i\rangle) \geq 0}(\langle w_r(t+1), x_i\rangle + \langle \mathsf{u}_r(t+1), x_i\rangle)| \\
&\leq |\langle \eta \Delta w_r(t), x_i\rangle + \langle \mathsf{u}_r(t), x_i\rangle - \langle \mathsf{u}_r(t+1), x_i\rangle| \\
&\leq U_{1,i,r} + U_{2,i,r}
\end{aligned}
$$

where the first step follows from Fact B.2, the fifth step follows from Definition C.8, and the last step follows from defining:

$$
\begin{aligned}
U_{1,i,r} &:= \langle \eta \Delta w_r(t), x_i\rangle \\
U_{2,i,r} &:= \langle \mathsf{u}_r(t), x_i\rangle - \langle \mathsf{u}_r(t+1), x_i\rangle
\end{aligned}
$$

For the first term $U_{1,i,r}$, we have:

$$|U_{1,i,r}| \leq \eta \frac{\sqrt{n}}{\sqrt{m}}\|\mathsf{F}(t) - y\|_2$$

this step holds since Part 2 of Lemma I.4.

For the second term $U_{2,i,r}$, we have:

$$|U_{2,i,r}| \leq O\left(\eta \frac{R^2 \sqrt{nd}}{\delta \epsilon \sqrt{m}} D\right)\|\mathsf{F}(t) - y\|_2$$

this step follows from Eq. (12) and Eq. (9).

Thus, we have:

$$
\begin{aligned}
C_4 &= \frac{1}{2}\|\mathsf{F}(t) - \mathsf{F}(t+1)\|_2^2 \\
&= \frac{1}{2}\sum_{i=1}^{n}(\mathsf{F}_i(t) - \mathsf{F}_i(t+1))^2 \\
&= \frac{1}{2}\sum_{i=1}^{n}\left(\kappa \frac{1}{\sqrt{m}}\sum_{r=1}^{m} a_r(U_{1,i,r} + U_{2,i,r})\right)^2
\end{aligned}
$$

Combining two terms, then by Hoeffding inequality (Lemma B.8), with a probability at least $1 - \delta$, $\mathbb{E}[\sum_{r=1}^{m} a_r(U_{1,i,r} + U_{2,i,r})] = 1$, we have:

$$|C_4| \leq O\left(\eta^2 \kappa^2 \frac{R^4 n^2 d}{\delta^2 \epsilon^2 m} D^2\right)\|\mathsf{F}(t) - y\|_2^2 \leq O\left(\eta^2 \kappa^2 \frac{R^4 n^2 d}{\delta^2 \epsilon^2 m} D^2\right)\mathsf{L}(t)$$

$\square$

# I Inductions

## I.1 Main Result 1: Training Convergence Guarantee

**Theorem I.1.** *If the following conditions hold:*

- *Let $D > 0$ be defined as Definition B.16.*

- *Given a expected error $\epsilon > 0$.*

- *Let $H(t) \in \mathbb{R}^{n \times n}$ be defined as Definition G.1.*

- *Let $H^* \in \mathbb{R}^{n \times n}$ be defined as Claim G.2. Assume $\lambda_{\min}(H^*) > 0$ as Assumption G.4.*

- *Let $\mathsf{L}(t)$ be defined as Definition C.9.*

- *Let $\mathsf{F}(t) \in \mathbb{R}^n$ be defined as Definition C.9.*

- *Let $\mathcal{D} = \{(x_i, y_i)\}_{i=1}^n \subset \mathbb{R}^d \times \mathbb{R}$ be defined as Definition C.1.*

- *Let $W(t) \in \mathbb{R}^{d \times m}$ be initialized as Definition C.3 and be updated by Definition C.8.*

- *$\delta \in (0, 0.1)$, $\kappa \in (0, 1]$.*

- *Choose $m \geq \Omega\left(\lambda^{-8} \frac{n^{12} d^8}{\delta^4 \epsilon^4}\right)$.*

- *Choose $\eta \leq O\left(\lambda \frac{\delta}{\kappa^2 n^2 dD}\right)$.*

- *Choose $T \geq \Omega\left(\frac{1}{\eta\lambda} \log(\epsilon^{-1} ndD^2)\right)$.*

*Then with probability at least $1 - \delta$, we have:*

$$\mathsf{L}(T) \leq \epsilon$$

*Proof.* **Choice of $m$.**

Following Lemma I.2, we have

$$m \geq \Omega\left(\lambda^{-4} \kappa^4 \frac{R^8 n^6 d^2}{\delta^4 \epsilon^4}\right)$$

Particularly, following Claim I.5, we have:

$$R \leq \frac{4\sqrt{n}}{\lambda\sqrt{m}} \|\mathsf{F}(0) - y\|_2$$

$$\leq \frac{4\sqrt{n}}{\lambda\sqrt{m}} \cdot O\left(\sqrt{n} dD^2\right)$$

$$\leq O\left(\frac{nd}{\lambda\sqrt{m}} D^2\right)$$

where the first step follows from Claim I.5, the second step follows from Lemma I.3, the third step follows from simple algebras.

Besides, by Lemma I.2, we need that

$$R \leq O\left(\frac{\lambda\delta}{\kappa^2 n^2 dD}\right)$$

where the second step follows from Definition B.16.

Thus, showing that $D^3 \leq O(m^{\frac{1}{4}})$ and $\kappa \leq 1$, we plug $m$ as follows:

$$m \geq \Omega\left(\lambda^{-8} \frac{n^{12} d^8}{\delta^4 \epsilon^4}\right)$$

**Choice of $\eta$.** We have

$$\|\eta \Delta w_r(0)\|_2 \leq \eta \frac{\sqrt{n}}{\sqrt{m}} \|\mathsf{F}(0) - y\|_2$$

$$\leq \eta \frac{\sqrt{n}}{\sqrt{m}} O\left(\sqrt{n} d D^2\right)$$

$$\leq R$$

where the first step follows from Part 2 of Lemma I.4, the second step follows from Lemma I.3, the third step follows from plugging $\eta \leq O\left(\lambda \frac{\delta}{\kappa n^2 d D}\right)$ and $m \geq \Omega\left(\lambda^{-8} \frac{n^{12} d^8}{\delta^4 \epsilon^4}\right)$.

**Choice of $T$.** We have:

$$\mathsf{L}(T) \leq \epsilon \iff (1 - \eta\lambda/2)^T \mathsf{L}(0) \leq \epsilon$$

$$\iff (1 - \eta\lambda/2)^T O\left(\sqrt{n} d D^2\right) \leq \epsilon$$

$$\iff (1 - \eta\lambda/2)^T \leq O\left(\frac{\epsilon}{\sqrt{n} d D^2}\right)$$

$$\iff T \geq \Omega\left(\log(\frac{\epsilon}{\sqrt{n} d D^2})/\log(1 - \eta\lambda/2)\right)$$

$$\iff T \geq \Omega\left(-\frac{1}{\eta\lambda}\log(\frac{\epsilon}{\sqrt{n} d D^2})\right)$$

$$\iff T \geq \Omega\left(\frac{1}{\eta\lambda}\log(\epsilon^{-1} n d D^2)\right)$$

where the first step follows from Lemma I.2, the second step follows from Lemma I.3, the third and fourth steps follow from simple algebras, the fifth step follows from Fact B.6, the sixth step follows from simple algebras. $\square$

## I.2 Induction for Loss

**Lemma I.2.** *If the following conditions hold:*

- *Let $D > 0$ be defined as Definition B.16.*

- *For $i, j \in [n]$, $r \in [m]$ and integer $t \geq 0$.*

- *Let $H(t) \in \mathbb{R}^{n \times n}$ be defined as Definition G.1.*

- *Let $H^\perp(t) \in \mathbb{R}^{n \times n}$ be defined as Definition G.3.*

- *Let $H^* \in \mathbb{R}^{n \times n}$ be defined as Claim G.2. Assume $\lambda_{\min}(H^*) > 0$ as Assumption G.4.*

- *Let $\mathsf{L}(t)$ be defined as Definition C.9.*

- *Let $\mathsf{F}(t) \in \mathbb{R}^n$ be defined as Definition C.9.*

- *Let $\mathcal{D} = \{(x_i, y_i)\}_{i=1}^n \subset \mathbb{R}^d \times \mathbb{R}$ be defined as Definition C.1.*

- *Let $W(t) \in \mathbb{R}^{d \times m}$ be initialized as Definition C.3 and be updated by Definition C.8.*

- *Let $a \in \mathbb{R}^m$ be initialized as Definition C.3.*

- *Let $\mathsf{dq} : \mathbb{R} \to \mathbb{R}$ be defined as Definition D.5.*

- *Denote $\widetilde{w}_r = \mathsf{q}(w_r) \in \{-1, +1\}^d$.*

- *Let $\mathcal{S}_i, \mathcal{S}_i^\perp$ be defined as Definition E.2.*

- *Let $\mathsf{u}_r(t)$ be defined as Definition H.1.*

- $\delta \in (0, 0.1)$.

- *For an error $\epsilon > 0$ and $\|\mathsf{F}(t) - y\|_2 \geq c \cdot \epsilon$ for a sufficient small constant $c > 0$.*

- $m \geq \Omega\left(\lambda^{-4}\kappa^4 \frac{R^8 n^6 d^2}{\delta^4 \epsilon^4}\right)$.

- $R \leq O(\frac{\lambda\delta}{\kappa^2 n^2 dD})$.

- *Define*

$$C_1 := -\kappa\frac{1}{\sqrt{m}}\sum_{i=1}^{n}\sum_{r \in \mathcal{S}_i} a_r \big(\mathbf{1}_{\mathsf{dq}(\langle \widetilde{w}_r(t), x_i\rangle) \geq 0}\langle w_r(t), x_i\rangle$$
$$- \mathbf{1}_{\mathsf{dq}(\langle \widetilde{w}_r(t+1), x_i\rangle) \geq 0}\langle w_r(t+1), x_i\rangle\big) \cdot (\mathsf{F}_i(t) - y_i)$$

- *Define*

$$C_2 := -\kappa\frac{1}{\sqrt{m}}\sum_{i=1}^{n}\sum_{r \in \mathcal{S}_i^{\perp}} a_r \Big(\mathbf{1}_{\mathsf{dq}(\langle \widetilde{w}_r(t), x_i\rangle) \geq 0}\langle w_r(t), x_i\rangle$$
$$- \mathbf{1}_{\mathsf{dq}(\langle \widetilde{w}_r(t+1), x_i\rangle) \geq 0}\langle w_r(t+1), x_i\rangle\Big) \cdot (\mathsf{F}_i(t) - y_i)$$

- *Define*

$$C_3 := -\kappa\frac{1}{\sqrt{m}}\sum_{i=1}^{n}\sum_{r=1}^{m} a_r \Big(\mathbf{1}_{\mathsf{dq}(\langle \widetilde{w}_r(t), x_i\rangle) \geq 0}\langle \mathsf{u}_r(t), x_i\rangle$$
$$- \mathbf{1}_{\mathsf{dq}(\langle \widetilde{w}_r(t+1), x_i\rangle) \geq 0}\langle \mathsf{u}_r(t+1), x_i\rangle\Big) \cdot (\mathsf{F}_i(t) - y_i)$$

- *Define*

$$C_4 := \frac{1}{2}\|\mathsf{F}(t) - \mathsf{F}(t+1)\|_2^2$$

- $\delta \in (0, 1]$.

*Then with probability at least $1 - \delta$, we have:*

$$\mathsf{L}(t+1) \leq (1 - \lambda/2\eta) \cdot \mathsf{L}(t)$$

*Moreover, we can show that:*

$$\mathsf{L}(t) \leq (1 - \lambda/2\eta)^t \cdot \mathsf{L}(0)$$

*Proof.* We have:

$$\mathsf{L}(t+1) \leq \mathsf{L}(t) + \Big(-\eta\lambda + O(\eta\frac{n^2 dRD}{\delta}) + O(\eta\kappa\frac{n^{1.5}dRD}{\delta})$$
$$+ O(\eta\kappa\frac{R^2 n^{1.5}\sqrt{d}}{\delta\epsilon\sqrt{m}}D) + O(\eta^2\kappa^2\frac{R^4 n^2 d}{\delta^2\epsilon^2 m}D^2)\Big) \cdot \mathsf{L}(t)$$
$$\leq \mathsf{L}(t) + \Big(-\eta\lambda + \frac{1}{8}\eta\lambda + \frac{1}{8}\eta\lambda + \frac{1}{8}\eta\lambda + \frac{1}{8}\eta\lambda\Big) \cdot \mathsf{L}(t)$$
$$\leq (1 - \eta\lambda/2)\mathsf{L}(t)$$

where the first step follows from Claim H.2, Lemma H.3, Lemma H.4, Lemma H.5, Lemma H.6 and $\eta\lambda \leq 1$, the second step follows from the choice of $R$ and $m$, the last step follows from simple algebras.

**Choice of $R$.** We have:

$$R \leq O(\frac{\lambda\delta}{\kappa^2 n^2 dD}) \tag{13}$$

where this step is following the combination of Lemma G.5 and $O(\eta\frac{\kappa^2 n^2 dRD}{\delta} \leq \frac{1}{8}\eta\lambda)$.

**Choice of $m$.** We have:

$$\sqrt{m} \geq \Omega\Big(\lambda^{-1}\kappa\frac{R^2 n^{1.5} d^{0.5}}{\delta\epsilon}D\Big)$$

$$\Longleftrightarrow \sqrt{m} \geq \Omega\Big(\lambda^{-1}\kappa\frac{R^2 n^{1.5} d^{0.5}}{\delta\epsilon}m^{\frac{1}{4}}\Big)$$

$$\Longleftrightarrow m^{\frac{1}{4}} \geq \Omega\Big(\lambda^{-1}\kappa\frac{R^2 n^{1.5} d^{0.5}}{\delta\epsilon}\Big)$$

$$\Longleftrightarrow m \geq \Omega\Big(\lambda^{-4}\kappa^4\frac{R^8 n^6 d^2}{\delta^4\epsilon^4}\Big)$$

where the first step follows from plugging $O(\eta\kappa\frac{R^2 n^{1.5}\sqrt{d}}{\delta\epsilon\sqrt{m}}D) \leq \frac{1}{8}\eta\lambda$, the last three steps follow from simple algebras. $\square$

**Lemma I.3.** *If the following conditions hold:*

- *Let $D > 0$ be defined as Definition B.16.*

- *For $i, j \in [n]$, $r \in [m]$ and integer $t \geq 0$.*

- *Let $\mathsf{L}(t)$ be defined as Definition C.9.*

- *Let $\mathsf{F}(t) \in \mathbb{R}^n$ be defined as Definition C.9.*

- *Let $\mathcal{D} = \{(x_i, y_i)\}_{i=1}^n \subset \mathbb{R}^d \times \mathbb{R}$ be defined as Definition C.1.*

- *Let $W(t) \in \mathbb{R}^{d \times m}$ be initialized as Definition C.3 and be updated by Definition C.8.*

- *Let $a \in \mathbb{R}^m$ be initialized as Definition C.3.*

- *Let $\mathsf{dq} : \mathbb{R} \to \mathbb{R}$ be defined as Definition D.5.*

- *Denote $\widetilde{w}_r = \mathsf{q}(w_r) \in \{-1, +1\}^d$.*

- *Let $\mathcal{S}_i, \mathcal{S}_i^\perp$ be defined as Definition E.2.*

- *Let $\mathsf{u}_r(t)$ be defined as Definition H.1.*

- *For an error $\epsilon > 0$ and $\|\mathsf{F}(t) - y\|_2 \geq c \cdot \epsilon$ for a sufficient small constant $c > 0$.*

*Then with probability at least $1 - \delta$, we have:*

$$\|\mathsf{F}(0) - y\|_2 \leq O\Big(\sqrt{n}dD^2\Big)$$

*Proof.* We have:

$$\|\mathsf{F}(0) - y\|_2 \leq \|\mathsf{F}(0)\|_2 + \|y\|_2$$
$$\leq \|\mathsf{F}(0)\|_2 + \sqrt{n}$$
$$\leq (\sum_{i=1}^n |\mathsf{F}_i(0)|^2)^{\frac{1}{2}} + \sqrt{n}$$
$$\leq (\sum_{i=1}^n |\kappa\frac{1}{\sqrt{m}}\sum_{r=1}^m a_r \cdot \mathsf{ReLU}\Big(\mathsf{dq}(\langle\widetilde{w}_r(0), x_i\rangle)\Big)|^2)^{\frac{1}{2}} + \sqrt{n}$$
$$\leq O\Big(\sqrt{n\log(m/\delta)}dD\Big) + \sqrt{n}$$

$$\leq O\left(\sqrt{n}dD^2\right)$$

where the first step follows from triangle inequality, the second step follows from $y_i \leq 1, \forall i \in [n]$ and simple algebras, the third step follows from the definition of $\ell_2$ norm, the fourth step follows from Definition C.9 and Definition C.5, the last two steps follow by Hoeffding's inequality (Lemma B.8), Definition C.1 and simple algebras, and we can show that:

$$\mathbb{E}[\sum_{r=1}^{m} a_r \cdot \mathsf{ReLU}\Big(\mathsf{dq}(\langle \widetilde{w}_r(0), x_i \rangle)\Big)] = 0$$

also,

$$\mathsf{dq}(\langle \widetilde{w}_r(0), x_i \rangle) = \sqrt{V(w_r(0))} \cdot \langle \widetilde{w}_r(0), x_i \rangle + E(w_r(0))\langle \mathbf{1}_d, x_i \rangle$$
$$\leq O(\sqrt{d}D) \cdot \sqrt{d} + O(D) \cdot \sqrt{d}$$
$$\leq O(dD)$$

where these steps follow from Definition D.5, Lemma I.6 and simple algebras. $\qquad \square$

## I.3  Induction for STE Gradient

**Lemma I.4.** *If the following conditions hold:*

- *For $i, j \in [n]$, $r \in [m]$ and integer $t \geq 0$.*
- *Let $\mathsf{L}(t)$ be defined as Definition C.9.*
- *Let $\mathsf{F}(t) \in \mathbb{R}^n$ be defined as Definition C.9.*
- *Let $\mathcal{D} = \{(x_i, y_i)\}_{i=1}^{n} \subset \mathbb{R}^d \times \mathbb{R}$ be defined as Definition C.1.*
- *Let $W(t) \in \mathbb{R}^{d \times m}$ be initialized as Definition C.3 and be updated by Definition C.8.*
- *Let $a \in \mathbb{R}^m$ be initialized as Definition C.3.*
- *Let $\mathsf{dq} : \mathbb{R} \rightarrow \mathbb{R}$ be defined as Definition D.5.*
- *Denote $\widetilde{w}_r = \mathsf{q}(w_r) \in \{-1, +1\}^d$.*
- *Let $\mathcal{S}_i, \mathcal{S}_i^{\perp}$ be defined as Definition E.2.*
- *Let $\mathsf{u}_r(t)$ be defined as Definition H.1.*
- *For an error $\epsilon > 0$ and $\|\mathsf{F}(t) - y\|_2 \geq c \cdot \epsilon$ for a sufficient small constant $c > 0$.*

*Then with probability at least $1 - \delta$, we have:*

- *Part 1. $\forall k \in [d]$*

$$|\Delta w_{r,k}(t)| \leq \sqrt{\frac{n}{m}} \cdot \|\mathsf{F}(t) - y\|_2$$

- *Part 2.*

$$\|\Delta w_r(t)\|_2 \leq \sqrt{\frac{n}{m}} \cdot \|\mathsf{F}(t) - y\|_2$$

*Proof.* **Proof of Part 1.** We have:

$$|\Delta w_{r,k}(t)| = |\kappa \frac{1}{\sqrt{m}} \sum_{i=1}^{n} a_r \cdot \mathbf{1}_{\mathsf{dq}(\langle \widetilde{w}_r(t), x_i \rangle) \geq 0} \cdot x_{i,k} \cdot (\mathsf{F}_i(t) - y_i)|$$

$$\leq \kappa \frac{1}{\sqrt{m}} \Big( \sum_{i=1}^{n} (a_r \cdot \mathbf{1}_{\mathsf{dq}(\langle \widetilde{w}_r(t), x_i \rangle) \geq 0} \cdot x_{i,k})^2 \Big)^{\frac{1}{2}} \cdot \|\mathsf{F}(t) - y\|_2$$

$$\leq \sqrt{\frac{n}{m}} \cdot \|\mathsf{F}(t) - y\|_2$$

where the first step follows from Definition F.2, the second step follows from Cauchy-Schwarz inequality, the third step follows from

$$\max_{r \in [m], i \in [n], k \in [d]} |\mathbf{1}_{\mathsf{dq}(\langle \widetilde{w}_r(t), x_i \rangle) \geq 0} \cdot x_{i,k}| \leq 1$$

the above equation follows from simple algebras and $\|x_i\|_i = 1$.

**Proof of Part 2.**

By $\|x\|_i = 1, \forall i \in [n]$, this proof is trivially the same as **Proof of Part 1**. $\qquad\square$

## I.4 Induction for Weights

**Claim I.5.** *If the following conditions hold:*

- *For $i, j \in [n]$, $r \in [m]$ and integer $t \geq 0$.*
- *Let $\mathsf{L}(t)$ be defined as Definition C.9.*
- *Let $\mathsf{F}(t) \in \mathbb{R}^n$ be defined as Definition C.9.*
- *Let $\mathcal{D} = \{(x_i, y_i)\}_{i=1}^{n} \subset \mathbb{R}^d \times \mathbb{R}$ be defined as Definition C.1.*
- *Let $W(t) \in \mathbb{R}^{d \times m}$ be initialized as Definition C.3 and be updated by Definition C.8.*
- *Let $a \in \mathbb{R}^m$ be initialized as Definition C.3.*
- *Let $\mathsf{dq} : \mathbb{R} \to \mathbb{R}$ be defined as Definition D.5.*
- *Denote $\widetilde{w}_r = \mathsf{q}(w_r) \in \{-1, +1\}^d$.*
- *Let $\mathcal{S}_i, \mathcal{S}_i^{\perp}$ be defined as Definition E.2.*
- *Let $\mathsf{u}_r(t)$ be defined as Definition H.1.*
- *For an error $\epsilon > 0$ and $\|\mathsf{F}(t) - y\|_2 \geq c \cdot \epsilon$ for a sufficient small constant $c > 0$.*

*Then with probability at least $1 - \delta$, we have:*

$$R := \max_{t \geq 0} \max_{r \in [m]} \|w_r(0) - w_r(t)\|_2 \leq \frac{4\sqrt{n}}{\lambda \sqrt{m}} \|\mathsf{F}(0) - y\|_2$$

*Proof.* We have

$$R = \max_{t \geq 0} \max_{r \in [m]} \|w_r(0) - w_r(t)\|_2$$

$$\leq \max_{t \geq 0} \max_{r \in [m]} \|\sum_{\tau=1}^{t} \eta \Delta w_r(\tau)\|_2$$

$$\leq \eta \max_{t \geq 0} \max_{r \in [m]} \sum_{\tau=1}^{t} \|\Delta w_r(\tau)\|_2$$

$$\leq \eta \frac{\sqrt{n}}{\sqrt{m}} \max_{t \geq 0} \sum_{\tau=1}^{t} \|\mathsf{F}(\tau) - y\|_2$$

$$\leq \eta \frac{\sqrt{n}}{\sqrt{m}} \max_{t \geq 0} \sum_{\tau=1}^{t} (1 - \eta \lambda / 2)^{\tau} \|\mathsf{F}(0) - y\|_2$$

$$\leq \frac{4\sqrt{n}}{\lambda\sqrt{m}} \|\mathsf{F}(0) - y\|_2$$

where the first step follows from the definition of $R$, the second step follows from Definition C.8, the third step follows from triangle inequality, the fourth step follows from Part 2 of Lemma I.4, the fifth step follows from Lemma I.2, the last step follows from Fact B.6. $\qquad\square$

**Lemma I.6.** *Let $\delta \in (0, 0.1)$. Let $D > 0$ be defined as Definition B.16. Let $E : \mathbb{R}^d \to \mathbb{R}$ be defined as Definition D.2. Let $V : \mathbb{R}^d \to \mathbb{R}$ be defined as Definition D.3. Let $W(0) \in \mathbb{R}^{d \times m}$ be initialized as Definition C.3, denote $W := [w_1, w_2, \cdots, w_m] \in \mathbb{R}^{d \times m}$ satisfying $\|w_r - w_r(0)\|_2 \leq R$ where $R \geq 0$, then with a probability at least $1 - \delta$, we have*

- *Part 1. $|w_{r,k}(0)| \leq O(D)$, $\forall r \in [m], k \in [d]$.*

- *Part 2. $\|w_r(0)\|_2 \leq O(\sqrt{d}D)$, $\forall r \in [m]$.*

- *Part 3. $\|w_r\|_2 \leq O(\sqrt{d}D + R)$, $\forall r \in [m]$.*

- *Part 4. $E(w_r(0)) \leq O(D)$, $\forall r \in [m]$.*

- *Part 5. $\sqrt{V(w_r(0))} \leq O(D)$, $\forall r \in [m]$.*

- *Part 6. $E(w_r) \leq O(D + R)$, $\forall r \in [m]$.*

- *Part 7. $\sqrt{V(w_r)} \leq O(D + R)$, $\forall r \in [m]$.*

*Proof.* This proof follows from the union bound of the Gaussian tail bound (Fact B.1) and some simple algebras. $\qquad\square$

# J Supplementary Setup for Classical Linear Regression

## J.1 Model Function

**Definition J.1.** *If the following conditions hold:*

- *For a input vector $x \in \mathbb{R}^d$.*

- *For a hidden-layer weights $W \in \mathbb{R}^{d \times m}$ as Definition C.2.*

- *For a output-layer weights $a \in \mathbb{R}^m$ as Definition C.2.*

- *Let $\mathsf{ReLU} : \mathbb{R} \to \mathbb{R}$ be defined as Definition C.4.*

- *Let $D = \{(x_i, y_i)\}_{i=1}^n \subset \mathbb{R}^d \times \mathbb{R}$ be defined as Definition C.1.*

- *$t \geq 0$, let $W(0) \in \mathbb{R}^{d \times m}$ and $a \in \mathbb{R}^m$ be initialized as Definition C.3.*

- *$W'(0) := W(0)$.*

- *Let $W'(t) \in \mathbb{R}^{d \times m}$ be updated as Claim J.3.*

- *$\kappa \in (0, 1]$.*

*We define:*

$$f'(x, W, a) := \kappa \frac{1}{\sqrt{m}} \sum_{r=1}^m a_r \cdot \mathsf{ReLU}(\langle w_r, x \rangle) \in \mathbb{R}$$

*Then we define the compact form of $f(x, W't), a)$, we define:*

$$\mathsf{F}'(t) = [f(x_1, W'(t), a), f(x_2, W'(t), a), \cdots, f(x_n, W't), a)]^\top \in \mathbb{R}^n$$

## J.2 Loss and Training

**Definition J.2.** *If the following conditions hold:*

- *Let $\mathcal{D} = \{(x_i, y_i)\}_{i=1}^n \subset \mathbb{R}^d \times \mathbb{R}$ be defined as Definition C.1.*
- *Let $W(0) \in \mathbb{R}^{d \times m}$ be initialized as Definition C.3.*
- *Let $a \in \mathbb{R}^m$ be initialized as Definition C.3.*
- *Let $f' : \mathbb{R}^d \times \mathbb{R}^{d \times m} \times \mathbb{R}^m \to \mathbb{R}$ be defined as Definition J.1.*
- *For any $t \geq 0$.*

*We define:*

$$\mathsf{L}'(t) := \frac{1}{2}\|\mathsf{F}'(t) - y\|_2^2$$

**Claim J.3.** *If the following conditions hold:*

- *Let $\mathcal{D} = \{(x_i, y_i)\}_{i=1}^n \subset \mathbb{R}^d \times \mathbb{R}$ be defined as Definition C.1.*
- *Let $W(0) \in \mathbb{R}^{d \times m}$ be initialized as Definition C.3.*
- *Let $f' : \mathbb{R}^d \times \mathbb{R}^{d \times m} \times \mathbb{R}^m \to \mathbb{R}$ be defined as Definition J.1.*
- *Let $\mathsf{L}'(t)$ be defined as Definition J.2.*
- *For any $t \geq 0$.*
- *Denote $\eta > 0$ aa the learning rate.*

*We define:*

$$W'(t + 1) := W'(t) - \eta \cdot \Delta W'(t)$$

*Here, we also define that:*

$$W'(t) := \frac{\mathrm{d}}{\mathrm{d}W'(t)}\mathsf{L}'(t)$$

$$= \sum_{i=1}^n (\mathsf{F}'_i(t) - y_i) \cdot \kappa \left[ a_1 \cdot \mathbf{1}_{\langle w'_1(t), x_i \rangle \geq 0} x_i \quad \cdots \quad a_m \cdot \mathbf{1}_{\langle w'_m(t), x_i \rangle \geq 0} x_i \right] \in \mathbb{R}^{d \times m}$$

*Proof.* This proof follows from simple algebras. $\square$

## J.3 Induction for Weights

**Lemma J.4** (See Corollary 4.1 and the fifth equation of page 6 in Du et al. [41])**.** *If the following conditions hold:*

- *$t \geq 0$, let $W(0) \in \mathbb{R}^{d \times m}$ and $a \in \mathbb{R}^m$ be initialized as Definition C.3.*
- *$W'(0) := W(0)$.*
- *Let $W'(t) \in \mathbb{R}^{d \times m}$ be updated as Claim J.3.*
- *$R \leq O(\frac{\lambda \delta}{\kappa^2 n^2 dD})$.*

*Then we have*

$$\|w'_r(t) - w'_r(0)\| \leq R$$

*Proof.* Following Corollary 4.1 in Du et al. [41], we can show that:

$$\|w'_r(t) - w'_r(0)\| \le \frac{4\sqrt{n}}{\sqrt{m}\lambda}\|\mathsf{F}'(0) - y\|_2$$

Then we can complete this proof by combining the equation above with Lemma J.5 and $R \le O(\frac{\lambda\delta}{n^2 dD})$ in Lemma conditions. $\square$

## J.4 Induction for Loss

**Lemma J.5.** *If the following conditions hold:*

- *Let $\mathcal{D} = \{(x_i, y_i)\}_{i=1}^{n} \subset \mathbb{R}^d \times \mathbb{R}$ be defined as Definition C.1.*
- *Let $W(0) \in \mathbb{R}^{d \times m}$ be initialized as Definition C.3.*
- *Let $a \in \mathbb{R}^m$ be initialized as Definition C.3.*
- *Let $f' : \mathbb{R}^d \times \mathbb{R}^{d \times m} \times \mathbb{R}^m \to \mathbb{R}$ be defined as Definition J.1.*
- *For any $t \ge 0$.*
- *$W'(0) := W(0)$.*
- *Let $W'(t) \in \mathbb{R}^{d \times m}$ be updated as Claim J.3.*
- *$\delta \in (0, 0.1)$.*

*Then with probability at least $1 - \delta$, we have:*

$$\|\mathsf{F}'(0) - y\|_2 \le O\Big(\sqrt{n}dD^2\Big)$$

*Proof.* We have:

$$
\begin{aligned}
\|\mathsf{F}'(0) - y\|_2 &\le \|\mathsf{F}'(0)\|_2 + \|y\|_2 \\
&\le \|\mathsf{F}'(0)\|_2 + \sqrt{n} \\
&\le (\sum_{i=1}^{n} |\mathsf{F}'_i(0)|^2)^{\frac{1}{2}} + \sqrt{n} \\
&\le (\sum_{i=1}^{n} |\kappa \frac{1}{\sqrt{m}} \sum_{r=1}^{m} a_r \cdot \mathsf{ReLU}\Big(\langle w'_r(0), x_i\rangle\Big)|^2)^{\frac{1}{2}} + \sqrt{n} \\
&= (\sum_{i=1}^{n} |\kappa \frac{1}{\sqrt{m}} \sum_{r=1}^{m} a_r \cdot \mathsf{ReLU}\Big(\langle w_r(0), x_i\rangle\Big)|^2)^{\frac{1}{2}} + \sqrt{n} \\
&\le O\Big(\sqrt{n\log(m/\delta)}dD\Big) + \sqrt{n} \\
&\le O\Big(\sqrt{n}dD^2\Big)
\end{aligned}
$$

where the first step follows from triangle inequality, the second step follows from $y_i \le 1, \forall i \in [n]$ and simple algebras, the third step follows from the definition of $\ell_2$ norm, the fourth step follows from Definition C.9 and Definition C.5, the fifth step follows from $W'(0) = W(0)$, the last two steps follow by Hoeffding's inequality (Lemma B.8), Definition C.1, $\kappa \le 1$ and simple algebras, and we can show that:

$$\mathbb{E}[\sum_{r=1}^{m} a_r \cdot \mathsf{ReLU}\Big(\langle w_r(0), x_i\rangle\Big)] = 0$$

also,

$$
\begin{aligned}
\langle w_r(0), x_i\rangle &= \langle w_r(0), x_i\rangle \\
&\le O(\sqrt{d}D) \le O(dD)
\end{aligned}
$$

where this step follows from Lemma I.6 and simple algebras. $\square$

# K Similarities

## K.1 Main Result 2: Training Similarity

**Theorem K.1.** *If the following conditions hold:*

- *Let $D > 0$ be defined as Definition B.16.*

- *Given a expected error $\epsilon > 0$.*

- *Let $H^* \in \mathbb{R}^{n \times n}$ be defined as Claim G.2. Assume $\lambda_{\min}(H^*) > 0$ as Assumption G.4.*

- *Let $\mathcal{D}_{\text{test}} := \{(x_{\text{test},i}, y_{\text{test},i})\}_{i=1}^n \subset \mathbb{R}^d \times \mathbb{R}$ be defined as Definition K.2.*

- *Let $\mathsf{F}'(t) \in \mathbb{R}^n$ be defined as Definition J.1.*

- *Let $\mathsf{F}(t) \in \mathbb{R}^n$ be defined as Definition C.9.*

- *Let $\mathsf{F}'_{\text{test}}(t) \in \mathbb{R}^n$ be defined as Definition K.3.*

- *Let $\mathsf{F}_{\text{test}}(t) \in \mathbb{R}^n$ be defined as Definition K.3.*

- *For any $t \geq 0$.*

- *Let $W(t) \in \mathbb{R}^{d \times m}$ be initialized as Definition C.3 and be updated by Definition C.8.*

- *$W'(0) := W(0)$.*

- *Let $W'(t) \in \mathbb{R}^{d \times m}$ be updated as Claim J.3.*

- *For any error $\epsilon_{\text{quant}} > 0$.*

- *$\delta \in (0, 0.1)$.*

- *Choose $\kappa \leq O(\frac{\epsilon_{\text{quant}}}{dD^2})$.*

*Then with probability at least $1 - \delta$, we have:*

- *Part 1. $|\mathsf{F}_{\text{test},i}(t) - \mathsf{F}'_{\text{test},i}(t)| \leq \epsilon_{\text{quant}}$.*

- *Part 2. $|\mathsf{F}_i(t) - \mathsf{F}'_i(t)| \leq \epsilon_{\text{quant}}$.*

*Proof.* **Proof of Part 1.** We have:

$$
\begin{aligned}
&|\mathbf{1}_{\mathsf{dq}(\langle \widetilde{w}_r(t), x_{\text{test},i}\rangle) \geq 0}(\langle w_r(t), x_{\text{test},i}\rangle + \langle \mathsf{u}_r(t), x_{\text{test},i}\rangle) \\
&\quad - \mathbf{1}_{\langle w'_r(t), x_{\text{test},i}\rangle \geq 0}\langle w'_r(t), x_{\text{test},i}\rangle| \\
&\leq |\langle w_r(t), x_{\text{test},i}\rangle + \langle \mathsf{u}_r(t), x_{\text{test},i}\rangle - \langle w'_r(t), x_{\text{test},i}\rangle| \\
&= |\langle w_r(0) - \eta \sum_{\tau=0}^{t-1} \Delta w_r(\tau), x_{\text{test},i}\rangle + \langle \mathsf{u}_r(t), x_{\text{test},i}\rangle - \langle w'_r(0) - \eta \sum_{\tau=0}^{t-1} \Delta w'_r(\tau), x_{\text{test},i}\rangle| \\
&= |-\langle \eta \sum_{\tau=0}^{t-1} \Delta w_r(\tau), x_{\text{test},i}\rangle + \langle \mathsf{u}_r(t), x_{\text{test},i}\rangle + \langle \eta \sum_{\tau=0}^{t-1} \Delta w'_r(\tau), x_{\text{test},i}\rangle| \\
&\leq |\langle \eta \sum_{\tau=0}^{t-1} \Delta w_r(\tau), x_{\text{test},i}\rangle| + |\langle \eta \sum_{\tau=0}^{t-1} \Delta w'_r(\tau), x_{\text{test},i}\rangle| + |\langle \mathsf{u}_r(t), x_{\text{test},i}\rangle| \\
&\leq R + R + |\langle \mathsf{u}_r(t), x_{\text{test},i}\rangle| \\
&\leq O\Big(d(D + R)\Big)
\end{aligned}
$$

where the first step follows from Fact B.2, the second step follows from Definition C.8 and Claim J.3, the third step follows from $w_r'(0) = w_r(0)$, the fourth step follows from triangle inequality, the fifth step follows from Claim I.5 and Lemma J.4, the last step follows from Lemma D.7 and $\delta \in (0, 0.1)$.

Then we have:

$$
\begin{aligned}
|\mathsf{F}_{\text{test},i}(t) - \mathsf{F}'_{\text{test},i}(t)| &\leq \Big| \kappa \frac{1}{\sqrt{m}} \sum_{r=1}^{m} a_r \Big( \mathbf{1}_{\mathsf{dq}(\langle \widetilde{w}_r(t), x_{\text{test},i} \rangle) \geq 0} (\langle w_r(t), x_{\text{test},i} \rangle + \langle \mathsf{u}_r(t), x_{\text{test},i} \rangle) \\
&\quad - \mathbf{1}_{\langle w_r'(t), x_{\text{test},i} \rangle \geq 0} \langle w_r'(t), x_{\text{test},i} \rangle \Big) \Big| \\
&\leq \kappa \sqrt{\log(m/\delta)} \cdot O\Big( d(D + R) \Big) \\
&\leq \epsilon_{\text{quant}}
\end{aligned}
$$

where the first step follows from Definition K.3, the second step follows from Hoeffding's inequality (Lemma B.8), $\mathbb{E}[\sum_{r=1}^{m} a_r \sigma_{i,r}] = 0$, $\sigma_{i,r} \leq O\Big( \frac{\sqrt{n}}{m}(D + R) + R/\delta \Big)$ and defining:

$$
\begin{aligned}
\sigma_{i,r} := |\mathbf{1}_{\mathsf{dq}(\langle \widetilde{w}_r(t), x_{\text{test},i} \rangle) \geq 0} (\langle w_r(t), x_{\text{test},i} \rangle + \langle \mathsf{u}_r(t), x_{\text{test},i} \rangle) \\
- \mathbf{1}_{\langle w_r'(t), x_{\text{test},i} \rangle \geq 0} \langle w_r'(t), x_{\text{test},i} \rangle|
\end{aligned}
$$

and the last step follows from choosing

$$
\kappa \leq O(\frac{\epsilon_{\text{quant}}}{dD^2 + dDR}) \leq O(\frac{\epsilon_{\text{quant}}}{dD^2})
$$

**Proof of Part 2.** This part can be proved in the same way as **Proof of Part 1.** $\qquad\square$

## K.2 Test Dataset for Generalization Evaluation

**Definition K.2.** *We define test dataset $\mathcal{D}_{\text{test}} := \{(x_{\text{test},i}, y_{\text{test},i})\}_{i=1}^{n} \subset \mathbb{R}^d \times \mathbb{R}$, where $\|x_{\text{test},i}\|_2 = 1$ and $y_{\text{test},i} \leq 1$ for any $i \in [n]$.*

**Definition K.3.** *If the following conditions hold:*

- *Let $\mathcal{D}_{\text{test}} := \{(x_{\text{test},i}, y_{\text{test},i})\}_{i=1}^{n} \subset \mathbb{R}^d \times \mathbb{R}$ be defined as Definition K.2.*
- *Let $f' : \mathbb{R}^d \times \mathbb{R}^{d \times m} \times \mathbb{R}^m \to \mathbb{R}$ be defined as Definition J.1.*
- *Let $f : \mathbb{R}^d \times \mathbb{R}^{d \times m} \times \mathbb{R}^m \to \mathbb{R}$ be defined as Definition C.5.*
- *For any $t \geq 0$.*
- *Let $W(t) \in \mathbb{R}^{d \times m}$ be initialized as Definition C.3 and be updated by Definition C.8.*
- *$W'(0) := W(0)$.*
- *Let $W'(t) \in \mathbb{R}^{d \times m}$ be updated as Claim J.3.*

*We define:*

$$
\begin{aligned}
\mathsf{F}'_{\text{test}}(t) &:= [f'(x_{\text{test},1}, W'(t), a), f'(x_{\text{test},2}, W'(t), a), \cdots, f'(x_{\text{test},n}, W'(t), a)]^{\top} \\
\mathsf{F}_{\text{test}}(t) &:= [f(x_{\text{test},1}, W(t), a), f(x_{\text{test},2}, W(t), a), \cdots, f(x_{\text{test},n}, W(t), a)]^{\top}
\end{aligned}
$$

## K.3 Function Similarity at Initialization

**Lemma K.4.** *If the following conditions hold:*

- *Let $D > 0$ be defined as Definition B.16.*
- *Let $\mathsf{q} : \mathbb{R}^d \to \{-1, +1\}^d$ be defined as Definition D.4.*

- *Let $E : \mathbb{R}^d \to \mathbb{R}$ be defined as Definition D.2.*

- *Let $V : \mathbb{R}^d \to \mathbb{R}$ be defined as Definition D.3.*

- *For a weight vector $w \in \mathbb{R}^d$.*

- *Denote quantized vector $\widetilde{w} := \mathsf{q}(w) \in \{-1, +1\}^d$.*

- *For a vector $x \in \mathbb{R}^d$ and $\|x\|_2 = 1$.*

- *Let $f' : \mathbb{R}^d \times \mathbb{R}^{d \times m} \times \mathbb{R}^m \to \mathbb{R}$ be defined as Definition J.1.*

- *Let $f : \mathbb{R}^d \times \mathbb{R}^{d \times m} \times \mathbb{R}^m \to \mathbb{R}$ be defined as Definition C.5.*

- *Let $W(0) \in \mathbb{R}^{d \times m}$ be initialized as Definition C.3.*

- *$W'(0) := W(0)$.*

- *$\delta \in (0, 0.1)$.*

- *For any error $\epsilon_{\mathrm{init}} > 0$.*

- *We choose $\kappa \leq O(\epsilon_{\mathrm{init}}/(\sqrt{d}D^2))$*

*Then with probability at least $1 - \delta$, we have:*

$$|f(x, W(0), a) - f'(x, W'(0), a)| \leq \epsilon_{\mathrm{init}}$$

*Proof.* We have:

$$
\begin{aligned}
&|\mathbf{1}_{\mathsf{dq}(\langle \widetilde{w}_r(0), x \rangle) \geq 0} \mathsf{dq}(\langle \widetilde{w}_r(0), x \rangle) \\
&\quad - \mathbf{1}_{\langle w_r(0), x \rangle \geq 0} \langle w_r(0), x \rangle| \\
&\leq |\mathsf{dq}(\langle \widetilde{w}_r(0), x \rangle) - \langle w_r(0), x \rangle| \\
&\leq |\sqrt{V(w_r(0))} \langle \widetilde{w}_r(0), x \rangle + E(w_r(0)) \cdot \langle \mathbf{1}_d, x \rangle - \langle w_r(0), x \rangle| \\
&\leq O(\sqrt{d}D)
\end{aligned}
$$

where the first step follows from Fact B.2, the second step follows from Definition D.5, the last step follows from Lemma I.6.

Then by Hoeffding inequality (Lemma B.8), with a probability at least $1 - \delta$, we have:

$$
\begin{aligned}
|f(x, W(0), a) - f'(x, W'(0), a)| &\leq \kappa |\frac{1}{\sqrt{m}} \sum_{r=1}^{m} a_r \widehat{\sigma}_r| \\
&\leq \kappa O(\sqrt{d}D) \cdot \sqrt{\log(m/\delta)} \\
&\leq O(\kappa \sqrt{d}D^2)
\end{aligned}
$$

where we have:

$$
\begin{aligned}
&\widehat{\sigma}_r := \mathbf{1}_{\mathsf{dq}(\langle \widetilde{w}_r(0), x \rangle) \geq 0} \mathsf{dq}(\langle \widetilde{w}_r(0), x \rangle) - \mathbf{1}_{\langle w_r(0), x \rangle \geq 0} \langle w_r(0), x \rangle \\
&\mathbb{E}[\sum_{r=1}^{m} a_r \widehat{\sigma}_r] = 1 \\
&|\widehat{\sigma}_r| \leq O(\sqrt{d}D)
\end{aligned}
$$

$\square$

