# OpenReview forum: "Unlock the Theory behind Scaling 1-bit Neural Networks"
_CPAL.cc/2025/Proceedings_Track — CPAL 2025 (Proceedings Track) Poster_

### Official Review · Reviewer_ZTwN · 2025-01-10

**Rating:** 6
**Confidence:** 3

**Review:**

### Overview

The paper develops a theoretical foundation for scaling laws in 1-bit neural networks, establishing how such networks converge under over-parameterization. The authors show that 1-bit networks can achieve near-equivalent performance to their full-precision counterparts.

### Strengths

- The paper rigorously proves a scaling law for 1-bit neural networks, offering theoretical guarantees for convergence and generalization in low-precision settings.
- The use of the NTK framework to analyze training dynamics in 1-bit models is robust and well-detailed, offering insights into optimization and generalization behavior.
- The paper is relatively clear.


### Weaknesses

- The experiments focus on synthetic mathematical functions, which may not fully represent the challenges of real-world tasks such as computer vision or NLP.

- While the theory supports scaling laws, the practical implications for multi-layer or Transformer architectures remain underexplored.

- The paper does not compare 1-bit networks directly with other quantization methods (e.g., 8-bit quantization) in terms of efficiency and performance trade-offs.

---

### Official Review · Reviewer_ubTu · 2025-01-14
**Review: A Rigorous Theoretical Foundation for Scaling 1-Bit Neural Networks**

**Rating:** 6
**Confidence:** 3

**Review:**

**Strengths**

1. This study provides a rigorous theoretical framework for understanding the scaling behavior of 1-bit neural networks using Neural Tangent Kernel (NTK) analysis.
2. This study conducts a thorough analysis of the "generalization difference" between 1-bit and full-precision networks, validating the practical use of 1-bit networks in real-world applications.

**Weaknesses**

1. The analysis primarily focuses on small MLPs and proposes extending the findings to deeper networks through Hierarchical Learning theory. However, it lacks rigorous proof for such extensions. For a scaling law study, empirical validation at larger scales is necessary.
2. The theoretical guarantees rely on several strong assumptions, such as the positive definite NTK matrix and specific initialization conditions. Additionally, careful tuning of the scaling coefficient is required. These factors may limit the generalization of the study's insights in practical implementations.
2. Although the paper discusses theoretical scaling properties, it does not provide a detailed analysis of practical computational benefits. Furthermore, it lacks concrete comparisons of memory usage and computational costs between 1-bit and full-precision models.

---

### Official Review · Reviewer_jAJh · 2025-01-14

**Rating:** 6
**Confidence:** 2

**Review:**

Strengths:
- This paper provides the first theoretical foundation for understanding scaling laws in 1-bit neural networks, filling a crucial gap in the literature.
- The paper provides comprehensive mathematical proofs and analysis.
- The results help explain why 1-bit quantization works well in practice

Weaknesses/Suggestions:
- The scope of the theoretical analysis is somewhat limited, focusing primarily on two-layer networks with ReLU activation. While the authors argue for extensibility to deeper networks via hierarchical learning, this could be more rigorously established.
- The analysis relies on several strong assumptions that may not always hold in practice, such as the positive definiteness of the kernel matrix. The required bounds on learning rates and model width could be impractical for very large models.
- The empirical evaluation focuses mainly on synthetic functions rather than real-world tasks.

---

### Meta-Review · Area_Chair_k9Dt · 2025-02-03

**Recommendation:** Accept (Poster)
**Confidence:** 3

**Metareview:**

This paper provides a theoretical work to understand scaling laws in 1-bit neural networks. According to reviewers, this paper provides a theoretical guarantee for convergence and generalization in low-precision settings. The analysis is in the well studied NTK framework, which offers insights into optimization and generalization behavior. My decision is acceptance with poster.

---

### Decision · Program_Chairs · 2025-02-11

Accept (Poster)